# What's the plan?
# Metrics for implicit planning in LLMs and their application to rhyme generation and question answering

**Jim Maar**
HPI / University of Potsdam
Germany
jim.maar@hpi-alumni.de

**Denis Paperno**
Utrecht University
Netherlands
d.paperno@uu.nl

**Callum McDougall, Neel Nanda**
Google DeepMind, London, UK

## Abstract

Prior work suggests that language models, while trained on next token prediction, show implicit planning behavior: they may select the next token in preparation to a predicted future token, such as a likely rhyming word, as supported by a prior qualitative study of Claude 3.5 Haiku using a cross-layer transcoder. We propose much simpler techniques for assessing implicit planning in language models. With case studies on rhyme poetry generation and question answering, we demonstrate that our methodology easily scales to many models. Across models, we find that the generated rhyme (e.g. "-ight") or answer to a question ("whale") can be manipulated by steering at the end of the preceding line with a vector, affecting the generation of intermediate tokens leading up to the rhyme or answer word. We show that implicit planning is a universal mechanism, present in smaller models than previously thought, starting from 1B parameters. Our methodology offers a widely applicable direct way to study implicit planning abilities of LLMs. More broadly, understanding planning abilities of language models can inform decisions in AI safety and control.

## 1 Introduction

Does the remarkable ability of modern language models in generating coherent text result from some form of implicit planning? Our paper develops methods for investigating this question, and applies them to the case studies of rhymed poetry and question answering.

In language production, humans are known to use various strategies to plan ahead for words that they utter (Ferreira & Swets, 2002; Lee et al., 2013; Huettig, 2015; Barthel et al., 2016). Similar mechanisms could plausibly apply in Transformer language models. In the context of text generation with language models, *planning* for goal token(s) is a process with two stages, forward and backward:

**Definition 1.1.** **Forward planning** is the creation of planning representations at earlier positions that encode properties of goal token(s) at later positions.

**Definition 1.2.** **Backward planning** is using those planning representations for generating intermediate tokens before the goal token(s) are produced.

Forward and backward planning are parts of the same overarching mechanism: the former creates a representation of the goal, and the latter conditions intermediate generation on that representation.

**Definition 1.3.** Planning is **implicit** if planning representations are part of hidden activations, and **explicit** if planning representations are model outputs.

**Definition 1.4.** Planning is **successful** when (i) forward planning creates representations that causally implicate the generation of goal tokens and (ii) backward planning produces intermediate tokens that increase the likelihood of goal token(s).

By its nature, *implicit* planning is harder to study than explicit planning, e.g. laying out the plan for a task in a chain of thought. To establish implicit planning behavior, we need evidence of both

successful forward planning and backward planning. In this study, we focus on phenomena where planning ahead for the kinds of words to produce is particularly transparent: rhyming, as suggested by the pioneering case study in Lindsey et al. (2025), and question answering.

Rhyming and question answering offer a unique window into planning, because it is linked to text elements (rhyming words or answer words) whose nature and position are predictable from general principles but not determined by immediately preceding tokens. In poetry, planning for a rhyme manifests as: (i) representations of the target rhyme family emerging at the end of the first line (forward planning), and (ii) intermediate tokens in the second line that facilitate reaching that rhyme (backward planning). In question answering, planning manifests as: (i) representations of the planned answer (e.g. *elephant*) emerging at the end of the question (forward planning), and (ii) intermediate tokens (e.g. *an*) in the answer that lead towards the planned answer (backward planning).

The contributions of our work include: (i) Datasets for the study of planning in rhyming (1050 lines from 10 rhyme families) and question answering (500 questions for 20 intended answer nouns). (ii) Quantitative metrics for assessing aspects of planning behavior: successful forward planning, backward planning, and successful backward planning. (iii) Evidence for implicit planning for the broadest range of language models to date, 23 diverse open weight language models (from 1B to 32B parameters). Our study uses localized steering intervention for both tasks. We show that mean activation difference steering works across the board, altering forward and backward planning. We find a lot of similarities, but also some differences between models we study (cf. Appendix H and M).

In what follows, section 2 provides the scientific context; section 3 explains the details of our methodology and experimental setup. Section 4 compares language models according to our metrics and discusses observations on the planning circuits, which parallel Lindsey et al.'s findings for Haiku. We summarize our work and discuss its limitations and further directions in section 5.

## 2 RELATED WORK

One piece of evidence for implicit planning in language models comes from the studies of unreliable chain of thought (CoT): "When we bias models toward incorrect answers, they frequently generate CoT explanations rationalizing those answers" Turpin et al. (2023); this also applies to newer reasoning models (Chen et al., 2025). When rationalizing, the model must be entertaining the suggested answer before the rationalizing CoT is generated. So a critical output can be implicitly planned long before it is produced, an instance of implicit *forward* planning. Furthermore, intermediate steps leading up to the answer (in this case, the chain of thought) are conditioned on the implicit plan, an instance of *backward* planning.

As a response to reports on LLMs producing useful plans in domains such as coding (Bairi et al., 2023, e.g.), skeptical arguments about the general planning ability of LLMs have been raised in the literature. Various studies (Valmeekam et al., 2023; Zhang et al., 2024; Stein et al., 2025) show empirically that explicit planning abilities of current LLMs are limited. The conceptual argument that "a system that takes constant time to produce the next token cannot possibly be doing principled reasoning on its own" (Kambhampati et al., 2024) naturally applies to implicit planning as well.

However, LLMs can plausibly engage in some forms of simpler, heuristic type of planning – just like humans, whose general planning capacities are limited (Kahneman & Tversky, 1977; Buehler et al., 2010), are known to still plan ahead in their speech production (Brown-Schmidt & Konopka, 2015). Indeed, shaping the rhythm and rhyme for the next poetic line is a task that can be solved well enough with limited planning capacities. This is supported by the success of previous poetry generation experiments that used relatively simple tools Hopkins & Kiela (2017); Lau et al. (2018); Ormazabal et al. (2022); Jhamtani et al. (2019); Ghazvininejad et al. (2016).

Implicit planning representations in LLMs have been found for specific models (Pochinkov et al., 2024; Men et al., 2024; Wu et al., 2024). In this paper, we aim to compare planning across a wider range of language models. One of our case studies is on rhymed poetry generation, inspired by Lindsey et al. (2025), the first study of implicit planning in poetry generation. The authors prompted Claude Haiku 3.5 with

> *Example* 1. A rhyming couplet:\n He saw a carrot and had to grab it\n

The model was able to complete the second line with the correct rhyme, e.g. *His hunger was like a starving rabbit*. Furthermore, elements of *implicit forward planning* mechanisms were identified: there are vector components in the activations of the token at the end of the first rhyming line (second \n above) that Lindsey et al. identify as representing planning for the rhyme of the next line. Some of these planning activation components, they argue, correspond to the potential rhyme *rabbit* and, if suppressed, lead to other rhyming words (such as *habit*) being generated much more often. It is even possible to inject a planning direction for a word such as *green* that does not fit the context, causing the model to produce it instead of a correct rhyme.

In addition, Lindsey et al. found evidence of *backward planning* in the rhyming context. The planning directions, they argue, influence the intermediate words. If *rabbit* is the dominant planned rhyme word, certain constructions are produced, e.g. comparison construction as in *His hunger was **like** a starving rabbit*. If *rabbit* planning directions are artificially suppressed, different intermediate parts of the line are generated, such as *His hunger was **a powerful habit***.

Lindsey et al.'s evidence is limited to a small number of persuasive examples and a single closed-source language model. Because of its proprietary nature and the technical complexity of the approach, Lindsey et al.'s findings cannot be independently reproduced. Our paper addresses these limitations. First, we assess forward and backward planning in both rhyming and question answering across a range of models. Second, we introduce several quantitative metrics based on a varied dataset of rhymes. Third, we address the complexity problem: the cross-layer transcoder (CLT) approach used by Lindsey et al., while offering diverse interpretability promises, is particularly complex, computationally expensive, and difficult to replicate. Just training a CLT for a single model of modest size requires days of compute on a highly performant and expensive GPU such as H100 (Ameisen et al., 2025, Appendix D).

As key intervention, we employ activation steering with vector addition, as in Turner et al. (2023). Directly manipulating model's activations, activation steering differs from other ways of controlling outputs such as prompting or different variants of controlled or constrained generation (Kumar et al., 2022; Lew et al., 2023; Loula et al., 2025). Unlike in constrained generation that can use phonetic constraints relevant for rhyming (Roush et al., 2022) or evolutionary optimization with rhyme scoring (Jobanputra et al., 2025), we do not constrain the model's outputs but modify the rhyming representations from which the language model generates the next line normally. As an implicit control mechanism, activation steering shares some conceptual similarity with classifier-free guidance in diffusion models (Ho & Salimans, 2022). However, unlike classifier-free guidance which operates during the iterative sampling process, our approach intervenes on a single position's representations in autoregressive LLM generation, making it more directly interpretable.

## 3 METHODOLOGY

Taking inspiration in the observations of Lindsey et al., we propose several metrics for quantifying implicit forward and backward planning, and apply them to a variety of language models. We propose simpler methods that do not involve the costly training and use of cross-layer transcoders, or other types of dictionary learning. We focus here on planning for a noun answer in question answering and for a rhyme family in poetry; see also Appendix H for a discussion of models planning to produce specific rhyming words.

**3.1 Data** In **rhyming** experiments, we call the set of words that all rhyme with each other a **rhyme family**. Words in a rhyme family tend to share a suffix, which we use to name the rhyme family. For example, the *-ing* rhyme family contains words such as *king* and *ring*. We chose 10 rhyme families and 20 pairs of rhyme families such that every rhyme family appears as the first in two pairs and the last in two pairs. For each rhyme family (RF), we generated 105 first lines using Claude 3.5 Sonnet. We split the lines into a train set $\mathbf{P}_{\mathrm{RF}}^{\mathbf{Train}}$ (85 lines) and test set $\mathbf{P}_{\mathrm{RF}}^{\mathbf{Test}}$ (20 lines). In practice, less data for estimating the steering vector was often enough, see Appendix B.

For **question answering** experiments, we created a dataset of 20 **noun pairs** where one noun begins with a vowel (requiring article "an") and the other with a consonant (requiring article "a"), such as (*eye*, *heart*). For each noun, we created 13 suggestive questions for training and 5 for testing, plus 7 neutral questions answerable by both nouns in a pair. We use notation $\mathbf{P}_{\mathrm{N}}^{\mathbf{Train}}$ and $\mathbf{P}_{\mathrm{N}}^{\mathbf{Test}}$ analogously to the rhyme family datasets. See Appendix A for complete dataset details for both tasks.

**3.2 Models** We test language models from four families (Gemma2, Gemma3, Qwen3 and Llama3.1/3.2) with model sizes between 1B and 32B parameters Team et al. (2024; 2025); Yang et al. (2025); Grattafiori et al. (2024). For each model we test both the base version and the instruction tuned versions.[1]

**3.3 Mean Activation Steering and Choosing a Steering Vector** If a model exhibits implicit forward planning, then an intervention on an early position (such as the newline in 1, or the last word of the first line) can alter the rhyme or the answer generated multiple tokens later.

We estimate a steering vector using the average activation difference like in Arditi et al. (2024). We calculate the average activation of one of the last positions of the prompt: the last newline (\n) token in the prompt (*newline steering*), the last word of before that (*last word steering*), or the question mark (question mark steering for questions). For example, in the rhyming context, we can subtract the average activation in lines that rhyme with *sick* from the average activation of the same position for lines that rhyme with *pain*. The resulting steering vector, when applied to the respective newline or last word token in the line *The house was built with sturdy, reddish brick \n*, can lead to 'rhymed' generations like *And stood for years, enduring wind and rain* instead of the unsteered baseline version *And stood for years, enduring every trick*.

Steering operates on a pair of prompt categories $(\mathbf{C_1}, \mathbf{C_2})$ such as rhyme families ($\mathbf{RF}$) or noun answers ($\mathbf{N}$). For each model, layer $l$, steering position, and prompt category pair $(\mathbf{C_1}, \mathbf{C_2})$ we extract a steering vector $s_{\mathbf{C_1} \to \mathbf{C_2}}^{(l,i)}$ by calculating the mean difference between the activations on their train sets

$$s_{\mathbf{C_1} \to \mathbf{C_2}}^{(l,i)} = m \cdot \left( \sum_{p \in \mathbf{P}_{\mathbf{C_2}}^{\mathbf{Train}}} \mathbf{x}_i^{(l)}(p) - \sum_{p \in \mathbf{P}_{\mathbf{C_1}}^{\mathbf{Train}}} \mathbf{x}_i^{(l)}(p) \right)$$

Here $m$ is a constant, set to 1.5,[2] $\mathbf{x}_k^{(l)}$ is the hidden activation of the model at layer $l$ and position $k$, and $i$ is either $\mathbf{pos}(\backslash \mathbf{n})$ (the position of the last newline token in $p$) or or $\mathbf{pos}(?)$ or the position of the last token before the newline or the question mark.

To apply a steering vector, we add it to the residual stream $\mathbf{x}_k^{(1)}$ on the correct layer and token position during generation. Note that **we apply the steering vector on one token only** (the last word, question mark or the newline token).

For the final steering vector of each model and category pair $s_{\mathbf{C_1} \to \mathbf{C_2}}$ we choose out of all layers and positions the combination where steering has the maximum intended effect.

While we opt for average activation difference steering because of its simplicity, it is also possible to obtain a steering vector with other methods, such as differences of classifier probe weights or SAE weights. For examples of generated outputs with alternative steering vectors, see Appendix I.

**3.4 Rhyming Metrics** Let $\mathbf{C_{RF}}$ be a collection of 1000 couplets generated by some model using the prompts in $\mathbf{P}_{\mathbf{RF}}^{\mathbf{Test}}$. We sample 50 generations per prompt $p \in \mathbf{P}_{\mathbf{RF}}^{\mathbf{Test}}$.

Let $\mathbf{C_{RF_1 \to RF_2}}$ be a collection of 1000 couplets generated by some model using the prompts in $\mathbf{P}_{\mathbf{RF}}^{\mathbf{Test}}$ while being steered with $s_{\mathbf{RF_1} \to \mathbf{RF_2}}$.

As evidence *successful forward planning*, we use effectiveness of the steering intervention: if at position X there was no planning representation for a later position Y, then intervening on position X would not have changed the outcomes for position Y in a predictable way. This is done by comparing the following two metrics.

**Fraction of Correct Rhyme Family.** We calculate the fraction of couplets in $\mathbf{C_{RF}}$, where the last word has the correct rhyme family (rhymes with the last word of the first). We made collections containing all words in each rhyme family to do this.

---

[1]Only Qwen3 32B did not have an open weight base model available.

[2]Mean activation difference estimation defines an inherent scale: steering vector can be used with multiplier of 1. While that often works, we found that slightly bigger values such as 1.5 or 2 produce a more consistent effect. Informally, imprecision in the estimation of the steering vector is compensated by increasing its magnitude.

**Fraction of Correct Rhyme Family (Steered).** This metric is calculated the same way as the Fraction of Correct Rhyme Family metric. We evaluate it separately for each rhyme family pair $(\mathbf{RF_1}, \mathbf{RF_2})$ on $\mathbf{C_{RF_1 \rightarrow RF_2}}$. The correct rhyme family is the one that was steered towards $(\mathbf{RF_2})$.

Successful steering means that planning representations can be manipulated successfully. This allows us to assess *backward planning* by measuring to what extent interventions on the planning activations affect the model's behaviors at intermediate positions. For example, Gemma2 9B model steered towards the *-ight* rhyme family will not just end the second line in 2 with a different word like *light* instead of *sing*, but will likely take a different path after *above*, leading to a more natural ending:

---

*Example* 2. Whispers of freedom found in a bird's wing\n Soaring above where true joy will sing
**Steered towards *-ight*:** Soaring above bathed in a golden light

---

The likelihood of different intermediate continuations due to steering is reflected in a shift in the probability distribution over the next token. To measure this effect, we propose the following *probability based metrics* discussed in detail in Appendix F.

We measure *successful backward planning* in poetry–to what extent words generated in the task's context leads up to the planned completion–using *regeneration metrics*. For example, we take the line *And stood for years, enduring every trick* and regenerate the last word without the original context that conditioned the rhyme. If backward planning involving the rhyming plan was used, such regeneration is expected to produce *trick* at a higher rate than for generated lines in other contexts, and at a higher rate than for lines generated in the same context but steering towards a different rhyme.

**Fraction of Correct Last Word Regeneration.** We extract the second lines of all the couplets in $\mathbf{C_{RF}}$. (So that there is no context that this is a rhyme). We also remove the last word in all of the second lines. Then we regenerate the last word using the resulting prompts and calculate the fraction of cases where the regenerated word is from the correct rhyme family $\mathbf{RF}$.

**Fraction of Correct Last Word Regeneration (Steered).** This metric is computed similarly to the Fraction of Correct Last Word Regeneration, but with couplets in $\mathbf{C_{RF_1 \rightarrow RF_2}}$, calculated as the fraction of cases where the regenerated word is from the target rhyme family $\mathbf{RF_2}$. We calculate this metric separately for each rhyme family pair $(\mathbf{RF_1}, \mathbf{RF_2})$ on $\mathbf{C_{RF_1 \rightarrow RF_2}}$.

### 3.5 Question answering metrics

**Fraction of Correct Answer** We calculate the fraction of answers in $\mathbf{C_N}$ which contain the intended answer word as a substring.

**Fraction of Correct Answer (Steered)** We calculate the fraction of answers which contain the intended answer word as a substring. We evaluate it separately for each noun pair $(\mathbf{N_1}, \mathbf{N_2})$ on $\mathbf{C_{N_1 \rightarrow N_2}}$. The correct answer is the one that was steered towards $(\mathbf{N_2})$.

**Fraction of *a/an*** We calculate the fraction of answers in $\mathbf{C_N}$ which contain the token *a* or *an*.

**Fraction of *a/an* (steered)** We calculate the fraction of answers which contain the token *a* or *an*. We evaluate it separately for each noun pair $(\mathbf{N_1}, \mathbf{N_2})$ on $\mathbf{C_{N_1 \rightarrow N_2}}$ to determine whether the fraction of *a* vs. *an* changes in the correct direction depending on $(\mathbf{N_2})$.

## 4 RESULTS

**4.1 Basic rhyming behavior** We observe that models differ in rhyming abilities. Generally, bigger models rhyme more consistently than smaller ones, and instruction-tuned models rhyme better than their base versions (Fig. 2, solid bars). These findings cannot be attributed to tokenization differences between models; see Appendix G for details.

In all models, we found evidence of successful backward planning from regeneration metrics: lines generated in the context of a rhyme are likely to be completed by a word of the intended rhyme family even without the original rhyming context. For all models, the metric is above chance (cf. Fig. 1.

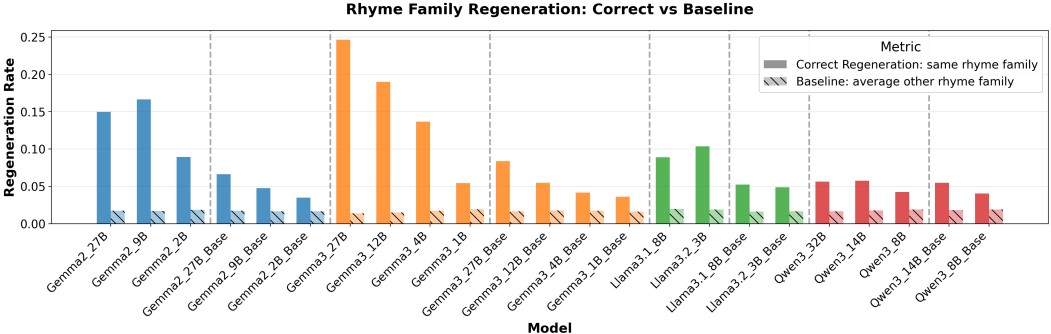

Figure 1: Comparison of unsteered regeneration rates to baseline chance level for the model. During regeneration with stochastic sampling, some rhyme families are expected to occur with non-zero frequency by chance. We estimate the baseline chance level that unsteered regeneration rate must exceed to show evidence for successful backward planning as the average frequency of rhyme families *other* than the one in whose context the couplet's second line was originally generated by the model.

**4.2 Steering affects forward and backward planning in rhyming** Steering on the last word consistently modifies models' behavior wrt the rhyme family generated. This supports that representations of the planned rhyme are localized at the intervention point. While rhyming abilities of different language models vary, our simple steering strategy achieved rates of the target rhyme family comparable to the rhyming rate of the model in the baseline condition, cf. Fig. 2. Only for models with the lowest rhyming ability (base variants of Gemma3 1B and Llama 3.2 3B) is the steered rhyming rate substantially lower than unsteered.

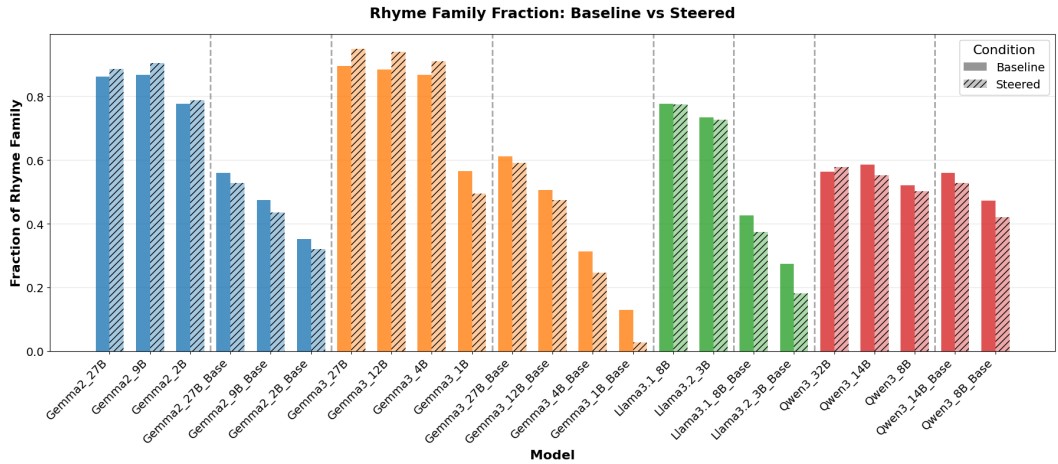

Figure 2: Baseline rhyming abilities of models vs. steered rhyming behavior. Solid bars report the frequency of (exact) rhymes produced by the model (baseline rhyming behavior). Hashed bars present the frequency of (exact) rhymes of rhyme family 2 after the first line ending in a word from rhyme family 1, when steered towards rhyme family 2.

Further, steered regeneration rates for the target rhyme family are close to the baseline regeneration rates for the same models (Fig. 3). This supports that the steering intervention does not just replace the final rhyming word but affects the backward planning that produces intermediate words.

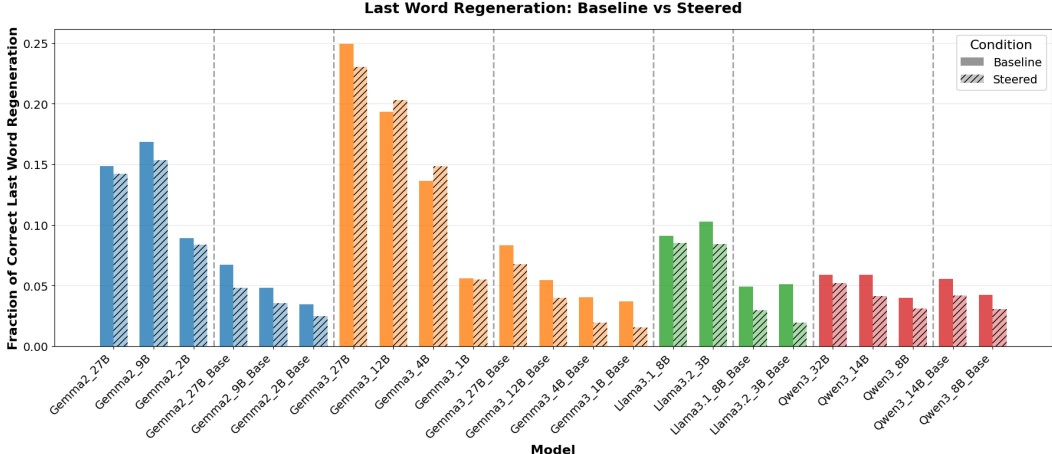

Figure 3: Baseline vs. steered last word regeneration rate of different models. Solid bars represent the fraction of times when regenerating the last word of the second line **with the first line removed** produces a word from the original rhyme family. This suggests that the model generates the second line so that it facilitates the correct rhyme, providing evidence for successful backward planning. Hashed bars represent the fraction of times when regenerating (with the first line removed) the last word of the second line **generated with steering towards rhyme family 2** produces a word from rhyme family 2. Close match between baseline and steered regeneration rates suggests that we successfully manipulate the LLM's rhyme planning.

The success of steering in affecting intermediate planning is supported by the fact that in models with high rhyming capabilities, in the baseline vs. steered lines, the regeneration frequency distributions by rhyme family is close, as illustrated in Fig. 4 for Gemma3 27B.

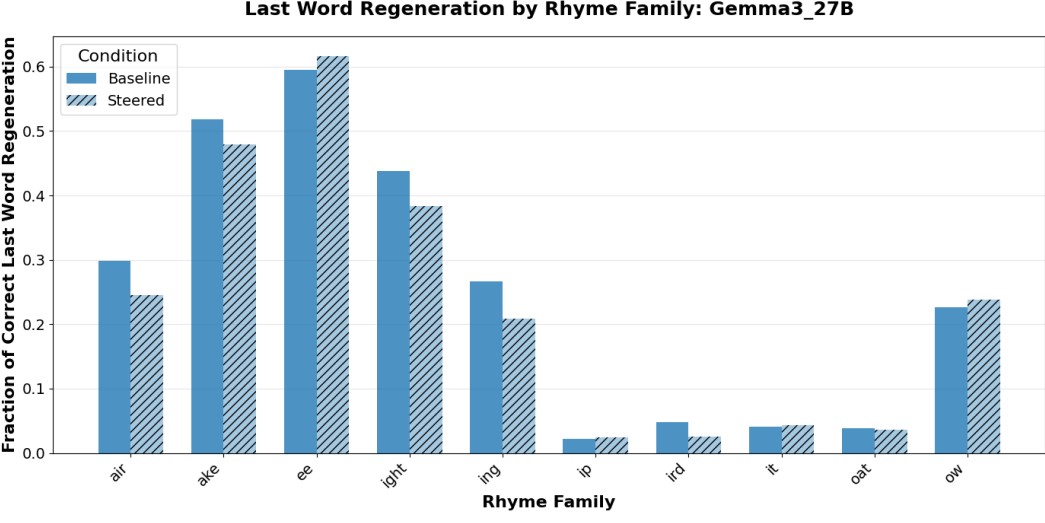

Figure 4: Regeneration rates per rhyme family with Gemma3 27B, baseline vs. steering. Solid bars represent the fraction of times when regenerating the last word of the second line **with the first line removed** produces a word from the original rhyme family. Hashed bars represent the fraction of times when regenerating (with the first line removed) the last word of the second line **generated with steering towards the rhyme family** produces a word from the target rhyme family. Close match between baseline and steered regeneration rates suggests that we successfully manipulate the LLM's rhyme planning for specific families.

**4.3   Probability based metrics**   Our probability based metrics detect evidence of backward planning on a finer level, assessing to what extent the probability distribution over intermediate tokens changes with this intervention. The probability based metrics generally follow the same patterns as the previous metrics: instruction tuned models tend to score higher than base models and bigger models tend to score higher than small ones. For plots, see (Fig. 14, 15) in Appendix F.

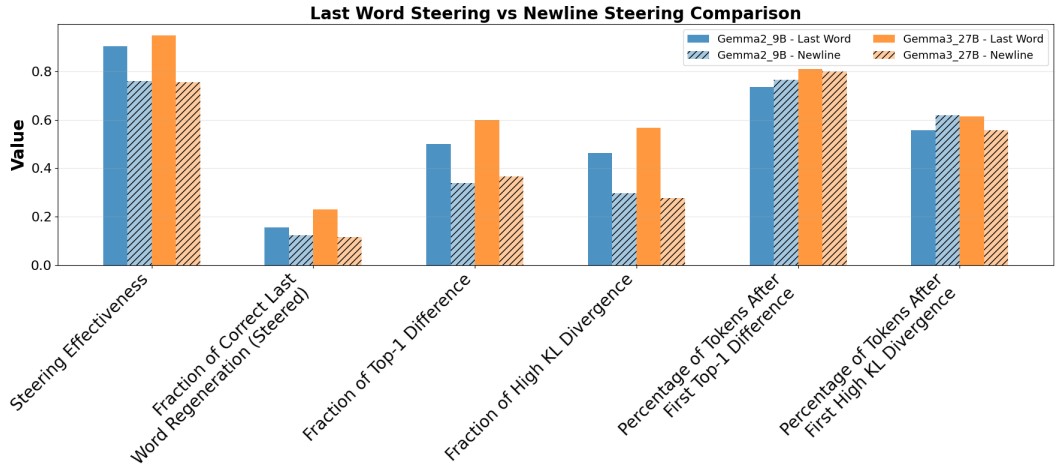

Figure 5: Steering on last word (solid bars) vs. newline token (hashed bars) at the end of the first line of a couplet. Both positions produce comparable metrics for Gemma2 9B and Gemma3 27B. Steering effectiveness is the fraction of times rhyme family 2 is generated when steering towards rhyme family 2; steering on the newline token is slightly less effective.

**4.4   Steering position in rhyming**   All models support rhyme family steering on the last word of the first line of a couplet, typically on lower layers. Some of the language models can also be steered on the newline token after the first line in one or more middle layers; for details, see Appendix M. The effect of newline steering is only pronounced for Gemma2 9B and Gemma3 27B (both instruction tuned and base variants), cf. Fig. 26 in the Appendix. While steering on the newline position is somewhat less effective than steering on the last word, it produces comparable values of planning metrics (Fig. 5). This suggests that interventions on the two positions are qualitatively similar. We conjecture the following explanation of quantitative differences: while the newline token may be playing a key role in the rhyming circuits of models like Gemma2 9B and Gemma3 27B, some amount of information flows around the steered newline token directly from the (unsteered) last word position to positions further in the sequence, dampening steering effectiveness on the newline position. This is supported by observations on attention patterns involved, see 4.6.

**4.5   Steering in Question Answering**   We tested whether our steering methodology for implicit planning generalizes to question answering using noun pairs requiring different articles (e.g., "an eye" vs. "a heart"). Steering vectors estimated from questions eliciting each noun successfully shifted answer frequencies, demonstrating forward planning. Critically, steering also affected article choice before the noun answer was generated: for example, models decreased "an" usage and increased "a" usage when steered toward a consonant-initial noun answer (Figure 6). This article selection, occurring before the answer itself, provides evidence of backward planning in question answering, consistent across all 23 models. On neutral questions, like in our rhyming study, instruction-tuned models showed a stronger steering effect. These findings suggest that implicit planning is a general mechanism rather than being specific to poetic constraints (full details in Appendix K).

We also conducted a small scale experiment with a more complex question answering task in which the strongest model in our study, Gemma3 27B, also shows signs of implicit planning multiple tokens ahead, with backward planning evidenced by subject-verb agreement. For details, see Appendix L.

**4.6   Observations on Planning Circuits**   How is backward planning implemented? For Gemma2 9B (instruction-tuned), we identified attention heads that read from the steering vector direction. For

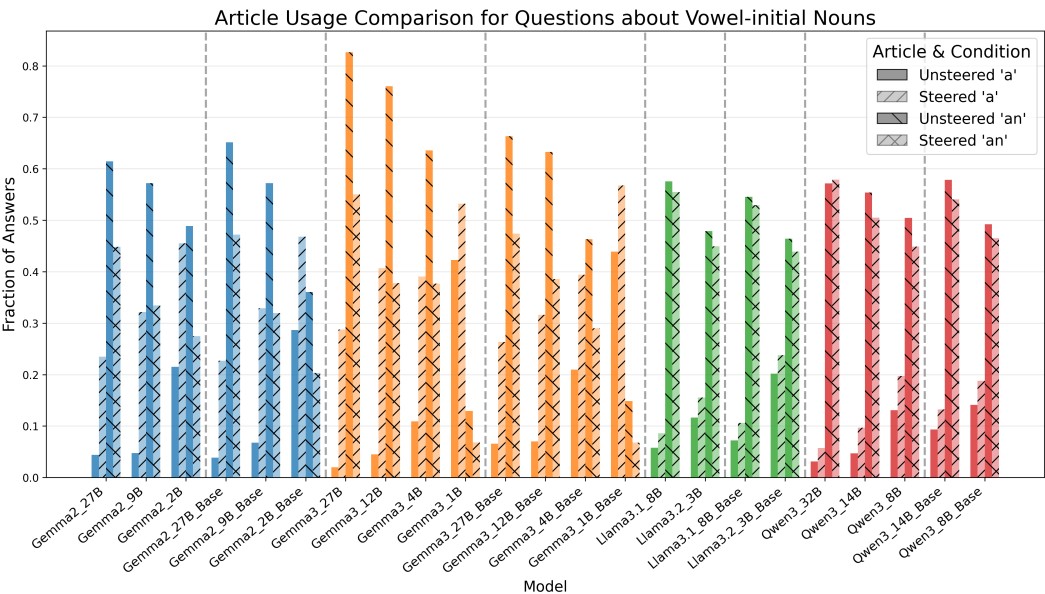

Figure 6: Percentage of answers containing article *a* or *an* as a function of steering on **suggestive questions**. Proportion of *an* decreases and proportion of *a* increases when steering towards a noun beginning with a consonant in a question about a noun beginning with a vowel.

| Couplet (unsteered completion)/(steered completion) | Logit Difference | | | % |
|---|---|---|---|---|
| | **Unst.** | **Patched** | **Steered** | |
| The house was built with sturdy, reddish brick
And stood for years, enduring *(every trick)/(sun and rain)* | -2.6 | 4.61 | 5.48 | 89 |
| He stubbed his toe, a striking pain
Now he's laid upon the floor, *(in vain)/(quite sick)* | -3.37 | 1.47 | 4.77 | 59 |
| Whispers of freedom found in a bird's wing
Soaring above
*(where true joy will sing)/(bathed in golden light)* | -14.5 | 2.06 | 3.39 | 93 |
| Mountains stand. As ancient guardians of majestic height
Echoing whispers of legends, though the
*(day and the night)/(the ages they sing)* | -4.94 | -1.07 | 0.93 | 66 |

Table 1: Couplets used for analyzing the rhyming circuit. The logit difference describes the difference in logits between the first tokens of the favored steered and unsteered completions. We look at three cases: **unsteered**, **steered**, and **patched**, in which we do not steer the model, but replace the activations of two attention heads (L30H03 and L31H15) with the activations they have under steering. **%** column indicates the percentage of steering induced logit difference recovered by patching L30H03 and L31H15. In all cases, most of the steering effect is recovered by patching the two attention heads, suggesting that they play a major role in rhyme plan implementation.

this analysis, we applied steering in the rhyming task on the newline token in layer 27. We analyzed positions with high KL divergence around the middle of the second line in examples steered between diverse rhyme pairs.

For example, the first row of table 1 the baseline model completes the second line with *every trick*. But when steering towards *-ain*, the favored completion is *sun and rain*.

Two attention heads (L30H3, L31H15) play a very important role in backward planning . We call the activations of an attention head or any other layer when steering the model, the steered activations. If

we do not use the steering vector, but instead replace the activations of these two attention heads on the last token with their steered activations (activation patching), we get a similar effect to steering. In the analyzed examples (table 1) activation patching results in token output logits that are much closer to the steered output logits then the unsteered output logits, recovering most of the steering effect (59%-93%). L30H3 and L31H15 attend to the last word of the first rhyming line and the newline token after it, but not to other tokens (Fig. 8 in Appendix). This attention pattern is constant across all the tokens of the second line, but only makes a significant contribution to the next token prediction at select positions, usually towards the end of the line. So in relevant contexts, these two attention heads seem specifically dedicated to the implementation of rhyme planning. The information copied by these heads is then converted into specific predictions in subsequent MLP layers 30–39; patching MLP layers recovers the effects of steering almost entirely. It remains an open question whether models apart from Gemma2 9B also have only a few attention heads which move rhyme planning information to later tokens.

In Gemma3 27B, which, like Gemma2 9B, also makes use of the newline token in rhyming, we similarly found that both the last word and the newline token of the first poetic line are attended to in generating the second line. To verify the causal role of this attention, we did an attention ablation study (Appendix D).

The attention heads that transfer rhyming information aren't contributing much in question answering, where most action happens in the later layers. In question answering, attention head L39H13 seems particularly important, with the final MLP layers, again, driving the final probability distribution prediction. This suggests that planning for a rhyme vs. answer to a question involve distinct circuits.

## 5 CONCLUSION

Our findings reveal that language models of various sizes exhibit different aspects of planning when generating rhymed poetry and answering questions. We find evidence of planning behavior even in the smallest models we consider, although it is weaker in smaller models, consistently with the conclusions of concurrent work (Hanna & Ameisen, 2026). Planning metrics increase not only with model size but also with instruction tuning, suggesting that typical post training may boost planning, cf. Li et al. (2024). The plan for a rhyme or an answer to a question can be manipulated using the simple technique of average activation difference steering. In poetry, this technique robustly recovers the model behavior both for the rhyming word generation and at the previous steps. Steering works reliably on the last word of the first rhyming line for all models, and in addition to that on the first line's newline token for select models, as in Claude Haiku, Gemma2 9B, and Gemma3 27B. It remains an open question for further exploration whether the more elaborate rhyming circuit in these models, which involves the newline token position, contributes to their quantitatively better planning behavior that we observed.

All our rhyming metrics correlate, suggesting that rhyming ability goes hand in hand with planning for a rhyme (see Appendix E for details). This holds for diverse logically independent aspects of planning: how early in the sentence the planning circuit springs to action (tokens after first top-1, tokens after first high KL); how much influence the planning circuit has over the output logits (fraction top-1 difference/high KL); how good the planning circuit is at boosting tokens leading up to the correct rhyme family (regeneration metrics).

Our steering experiments support that the planned rhyme family is represented at the end of the first line in a couplet and the planned asnwer is represented at the end of a question. The follow up analysis of Gemma2 9B further identifies attention heads and MLP layers that contribute the most to *implementing* the rhyming plan. This circuit is specific to the task; question answering uses circuits of a similar shape, but different in details. Our findings are consistent with known observations on rhyming circuits in other models (Lindsey et al., 2025; Hanna & Ameisen, 2026), suggesting a general mechanism, although details such as which positions are used differ between models.

Methods we developed can be transferred from rhyming and question answering to other cases where long-distance dependencies can be manipulated, such as instruction following and CoT (Cox, 2025), as well as to other intervention methods. Our findings call for further analysis of LLM planning. Since implicit planning is pervasive, especially in larger, more capable models, and may have critical safety consequences in certain domains, we need to better understand the computation involved.

ACKNOWLEDGEMENTS

The first exploratory steps of the research reported in this paper were taken as part of the training phase of Neel Nanda's stream at MATS program. The compute costs of the main body of work was covered by funds from Manifold for Charity. Denis Paperno was supported by Institute for Language Sciences at Utrecht University. We thank Michael Hanna for helpful comments on an earlier version of this paper.

REPRODUCIBILITY STATEMENT

Supplementary material on OpenReview contains data reported in the body of the paper and code for reproducing the experiments. This includes outputs of the rhyming experiments in `rhyme_family_steering` and outputs of question answering experiments in `noun_qa_steering`. Supplementary material including code, data and outputs of the both rhyming and question answering experiments is also found at `https://github.com/dpaperno/implicit-planning-supplementary-material.git`. README.MD contains further information needed for reproducing the experiments.

IMPACT STATEMENT

"This paper presents work whose goal is to advance the field of Machine Learning. There are many potential societal consequences of our work, none which we feel must be specifically highlighted here."

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

# A  DATASET DETAILS

## A.1  RHYMING DATASET

For our rhyming evaluation we steered between the following 20 rhyme family pairs: (-ing, -air), (-ing, -ip), (-air, -ip), (-air, -oat), (-ip, -oat), (-ip, -ird), (-oat, -ird), (-oat, -ee), (-ird, -ee), (-ird, -ight), (-ee, -ight), (-ee, -ake), (-ight, -ake), (-ight, -ow), (-ake, -ow), (-ake, -it), (-ow, -it), (-ow, -ing), (-it, -ing), (-it, -air). For each rhyme family $\mathbf{RF}_1$, 105 first lines ending with words from $\mathbf{RF}_1$ were generated using Claude 3.5 Sonnet. The resulting 105 lines per family were randomly split into 85 training and 20 test prompt lines. We then manually checked the training data, replacing individual lines with incorrect rhyming words with newly generated lines. For some rhyme families we also replaced the whole training data if it was heavily unbalanced, with a large share of examples ending in the same word. In case of complete replacement of lines for a rhyme family, we prompted Claude 4.0 to list 17 words belonging to the rhyme family, and after manually checking their correctness, to produce 5 poetry lines ending in each word. When developing our methods, we found that balanced training data improves the performance of the estimated steering vector.

For steering vector estimation and inference, each first line was prepended with `"A rhyming couplet:\n"`.

## A.2  NOUN QUESTION ANSWERING DATASET

The noun question answering study was based on the following 20 noun pairs: ('eye', 'heart'), ('ear', 'brain'), ('elephant', 'whale'), ('owl', 'shark'), ('actor', 'doctor'), ('athlete', 'student'), ('airport', 'hospital'), ('office', 'school'), ('ocean', 'river'), ('arrow', 'star'), ('heart', 'eye'), ('brain', 'ear'), ('whale', 'elephant'), ('shark', 'owl'), ('doctor', 'actor'), ('student', 'athlete'), ('hospital', 'airport'), ('school', 'office'), ('river', 'ocean'), ('star', 'arrow').

The data was selected from an initial set of 100 nouns. We used Claude Opus 4.1 to generate 30 diverse suggestive questions about each noun. We then tested three small models: Gemma3 4B, LLama3.2 8B, and Qwen3 8B on the questions. We selected easy, highly suggestive questions by retaining only questions that at least 2 out of the three small models answered correctly. Finally, we paired nouns that had at least 18 retained questions into pairs where one noun starts with a vowel and the other starts with a consonant and the nouns in a pair represent the same ontological category.

The neutral questions were generated by Claude Opus 4.1 so that each question can be somewhat plausibly answered by both nouns in the pair, e.g. *What large mammal is endangered?* for (*elephant*,*whale*).

For steering vector estimation and inference, each question was prepended with the 2-shot context `"Question: What two-wheeled vehicle do you pedal?\nAnswer: a bicycle\n\nQuestion: What flying vehicle carries passengers in the sky?\nAnswer: an airplane\n\nQuestion:"` and postpended with `"\nAnswer:"`.

# B  TRAINING DATA SIZE

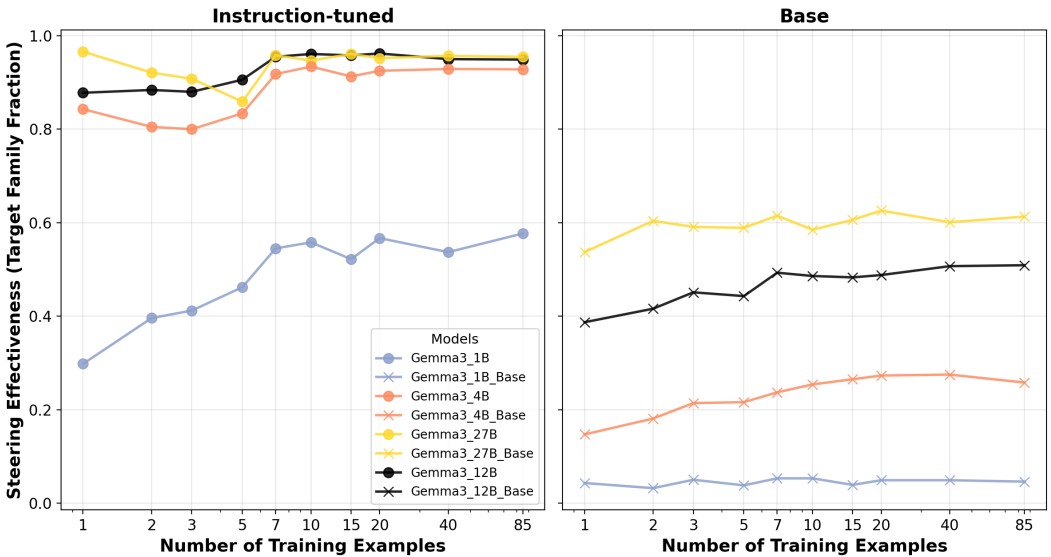

Figure 7: Steering effectiveness vs. training size data for Gemma 3 models.

We found that additional training examples mostly benefit steering effectiveness in less capable models, while the most capable models can generalize well from a just a few examples or even a single example pair. Figure 7 illustrates the efficiency of steering as a function of the number of training examples used to estimate the steering vector for Gemma3 models. Results are reported for the best layer and position for steering. For a fair comparison of models, we used a large set of 85 training examples for each rhyme family.

# C ATTENTION PATTERNS

In Gemma-2 9B, which allows for efficient newline token steering, attention heads which read from the steering vector attend both to the last word and the newline token, see Fig. 8. The rhyming circuit therefore crucially involves both tokens; the newline token, while similar in function to what Lindsey et al. report for Claude Haiku, does not play an exclusive role. Note that Lindsey et al. do not analyze activations on the last word of the first line.

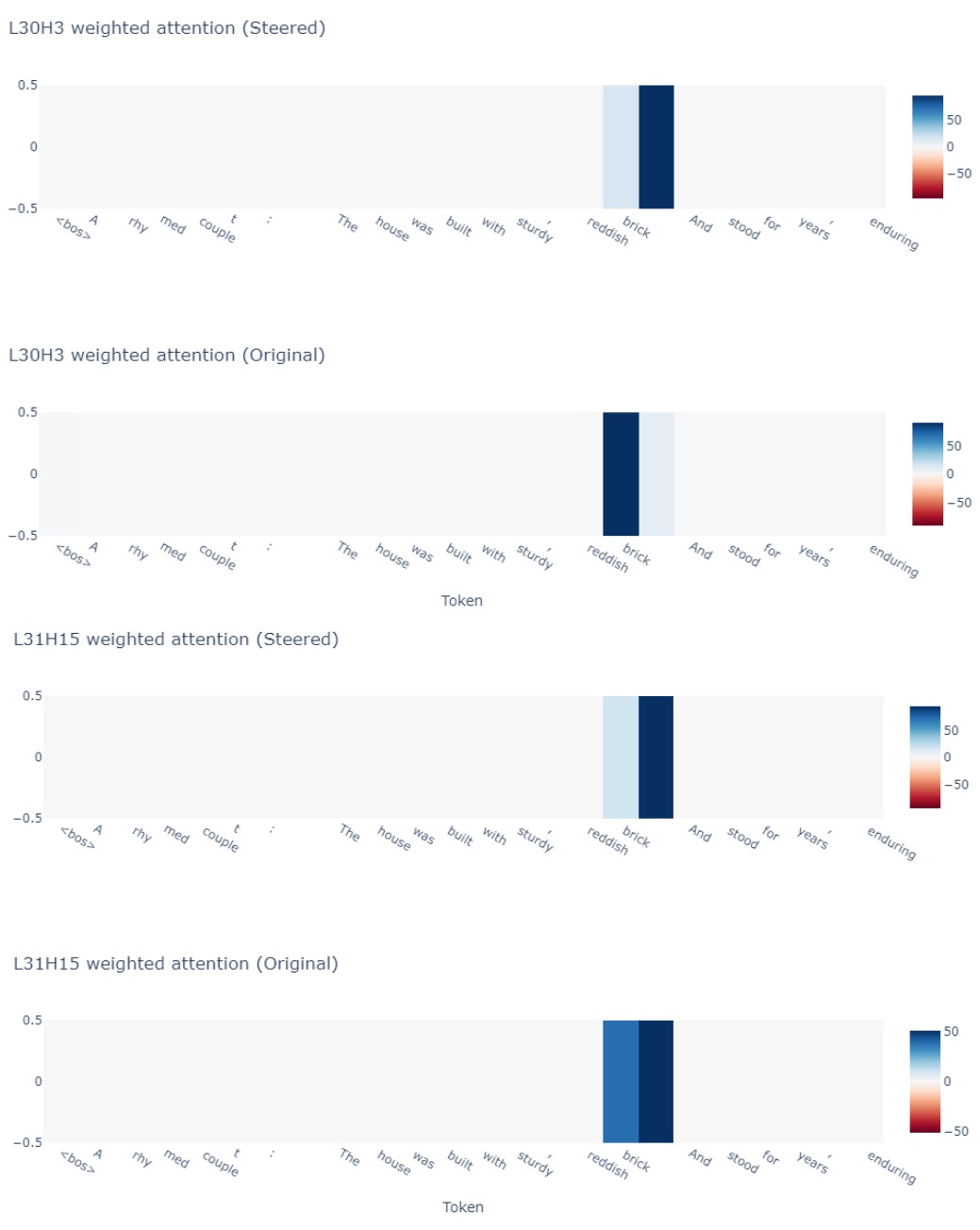

Figure 8: Attention patterns for heads L30H3 and L31H15 at a fork point.

# D   ATTENTION ABLATION

To strengthen our results we also do an attention ablation study. On Gemma3 27B, we measure it's ability to complete the rhyme and plan for the last word using the Fraction of correct rhyme family and Fraction of correct regeneration metrics. We do so in 4 different conditions, shown in Figure 9, in which the attention mask of the model is modified such that there is no attention paid to specific tokens and therefore also no information flow from these tokens to later positions.

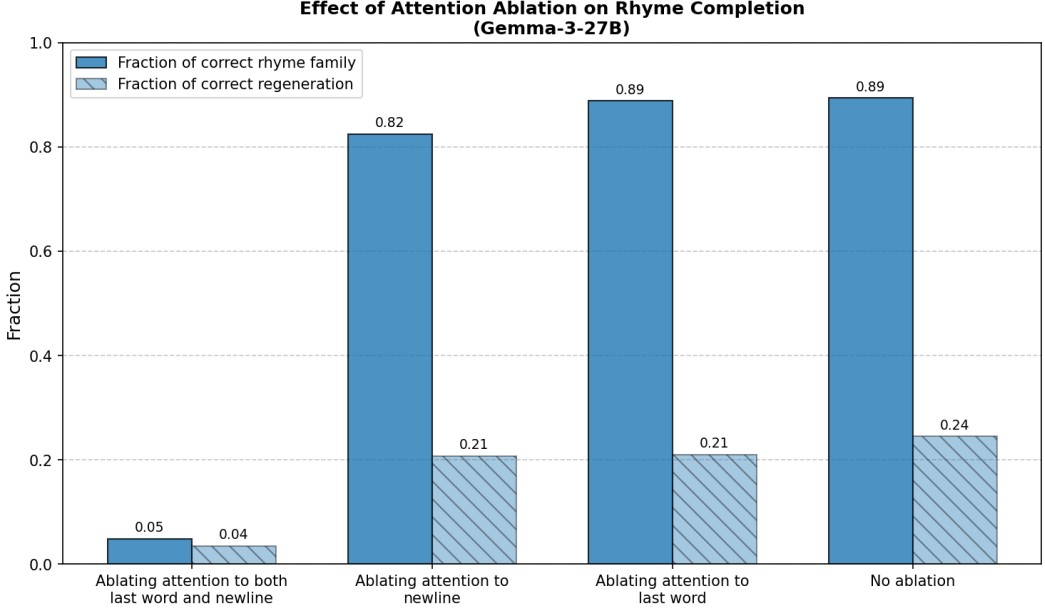

Figure 9: Fraction of correct rhyme family and Fraction of correct regeneration on the rhyming dataset for Gemma3 27B under 4 different conditions. Attention ablation of the last word of the first line and the newline token, attention ablation of the last word, attention ablation of the newline token and the baseline condition without ablation.

We observe that, as expected, the model is not able to complete the task well when attention to both the last word and the newline token is ablated. When only the attention to one of those is ablated, the model is able to plan well, but the highest regeneration scores are only reached without ablation. Like our previous analysis, this suggests that representations on both tokens are relevant for the backward planning ability of the model in the context of poetry.

# E CORRELATIONS OF DIFFERENT METRICS

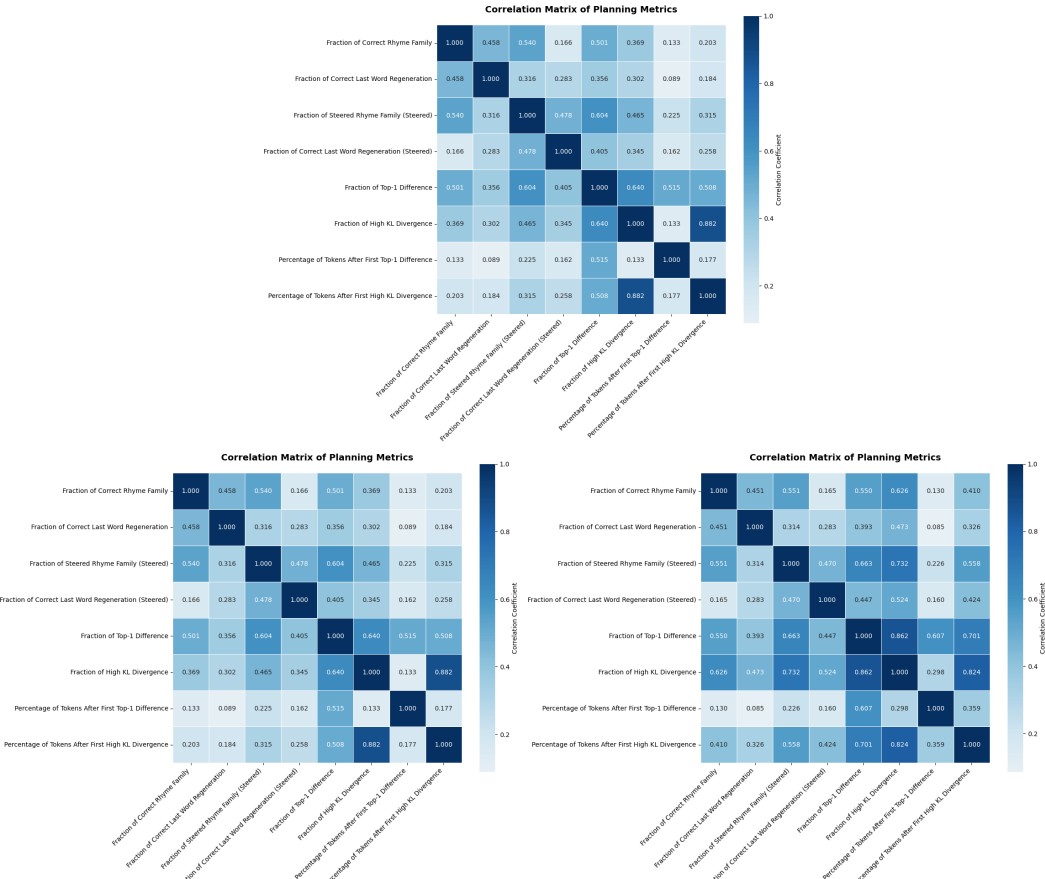

Figure 10: Correlations of different metrics at individual prompt level; all models vs. with exclusion of Gemma 2 base models, which tend to show idiosyncratically high KL divergence values.

We report correlations of all our metrics for multiple settings. This includes correlations of metrics for (prompt,model) pairs, as well as correlations between models with metrics averaged across all prompts.

All of our rhyming metrics correlate with rhyming correctly (Fig. 10), but they measure different aspects of planning in practice. If we control the data to only include instances where the model can produce a correct rhyme with high probability, metrics' correlations start to diverge. 'Tokens after' metrics become especially independent; see Fig. 12. In other words, badly rhymed outputs can be associated with poor involvement of rhyming circuits. On the other hand, provided that the model's rhyming behavior is stable, earlier execution of the rhyming plan does not necessarily lead to better rhyming; we have seen examples where an appropriate rhyming word is generated too early, before the line would naturally end.

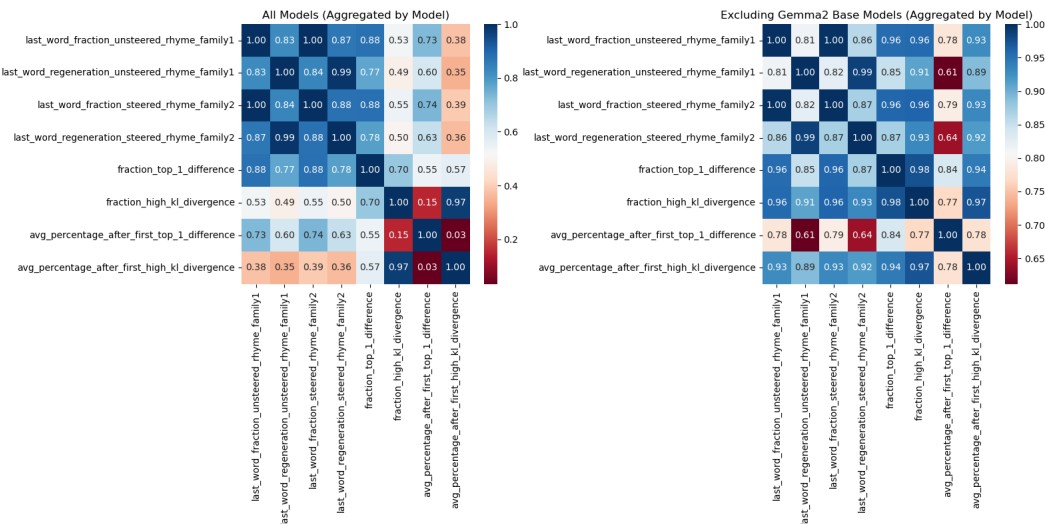

Figure 11: Correlations of different metrics at model level; all models vs. with exclusion of Gemma 2 base models, which tend to show idiosyncratically high KL divergence values.

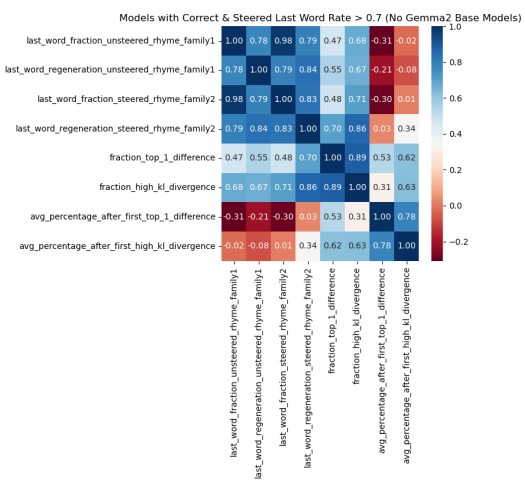

Figure 12: Correlations of different metrics, data filtered for reliable baseline and steered rhymes, aggregating data by model. Excluding Gemma2 Base models, which tend to show idiosyncratically high KL divergence values.

## F  PROBABILITY BASED METRICS IN RHYMING

Let $\mathbf{Y_{RF}}$ be a collection of sequences of probability distributions over text tokens generated by using $\mathbf{C_{RF}}$ as input to a model.

Let $\mathbf{Y_{RF_1 \rightarrow RF_2}}$ be a collection of sequences of probability distributions over text tokens generated by using $\mathbf{C_{RF_1}}$ as input to a model while steering with $s_{\mathbf{RF_1 \rightarrow RF_2}}$.

**Fraction of Top-1 Difference.**  Divergence in the top 1 most probable next token predictions of the steered and unsteered models signals backward planning in action. We calculate the fraction of such divergent positions in the second line of a couplet.

**Fraction of High KL Divergence.**  We calculate the average fraction of tokens in the second line where the KL divergence between the next token probability distribution using the steered vs.

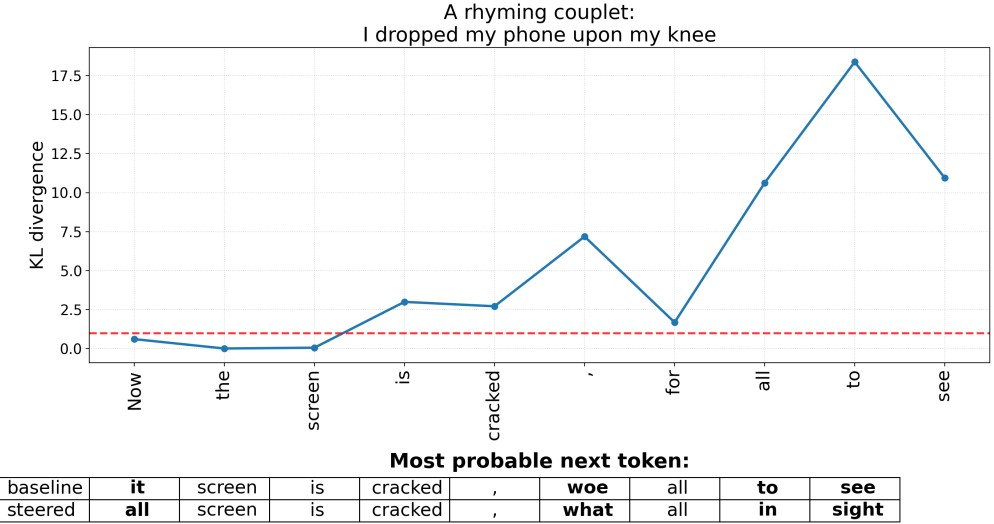

| | | | | | | | | | |
|---|---|---|---|---|---|---|---|---|---|
| **Most probable next token:** | | | | | | | | | |
| baseline | **it** | screen | is | cracked | , | **woe** | all | **to** | **see** |
| steered | **all** | screen | is | cracked | , | **what** | all | **in** | **sight** |

Figure 13: Illustration of the probability based metrics. Gemma3 27B model, baseline vs. steering from *-ee* to *-ight* rhyme family. Dashed line = KL divergence threshold of 1; in the table, **bold** = different top-1 prediction under steering. At the first token *Now*, the baseline and steered runs' probability distributions are still similar (KL divergence under 1), but the top tokens are different (*it* for baseline, *all* for steered). At the comma (,), not only are the top candidate tokens different, but KL divergence is also above 1.

unsteered model is greater than 1, out of all tokens in the second line of the couplet. Formula:

$$\frac{1}{|\mathbf{Y}_{\mathbf{RF_1}}|} \cdot \sum_{i \in [|\mathbf{Y}_{\mathbf{RF_1}}|]} \frac{1}{|\mathbf{sl}(\mathbf{C}_{\mathbf{RF}}[i])|} \sum_{j \in \mathbf{sl}(\mathbf{C}_{\mathbf{RF}}[i])} \mathbb{1}_{\mathbf{KL\ Divergence}(\mathbf{Y}_{\mathbf{RF_1}}[i,j], \mathbf{Y}_{\mathbf{RF_1} \rightarrow \mathbf{RF_2}}[i,j]) > 1}$$

**Fraction of Tokens After First Top-1 Difference.** We calculate the average position where the first top-1 difference occurs, measured in % of tokens of the second couplet line, counted from the end of the line. This measures how early backward planning kicks in on average.

**Fraction of Tokens After First High KL Divergence.** We calculate the average position where the first high KL divergence occurs, measured in % of tokens of the second couplet line, counted from the end of the line. This is another measure of how early backward planning kicks in on average.

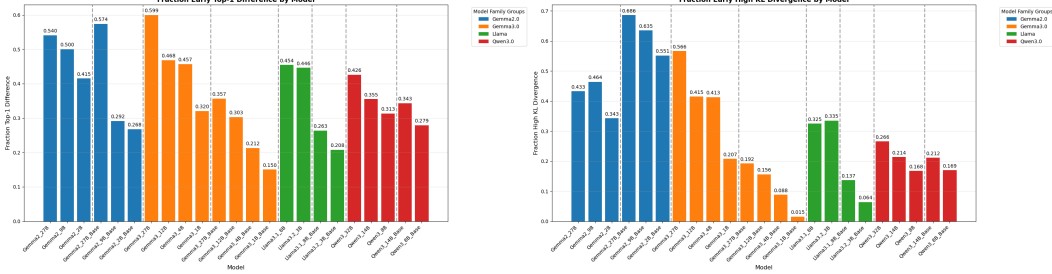

Figure 14: Percentage of top-1 difference and high KL divergence under steering for different models; higher percentage indicates stronger backward planning.

Metrics do show some idiosyncrasies. Perhaps due to the threshold for high KL divergence interacting with model properties, base versions of Gemma2 models have elevated KL divergence scores (as seen especially in Fig. 14, right), even if it does not correspond to elevated values of non-KL metrics.

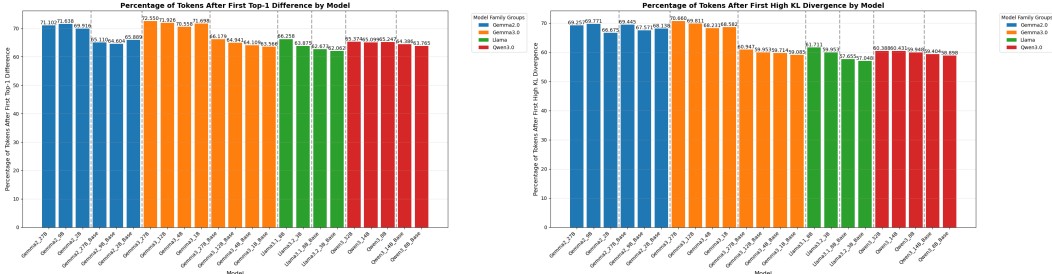

Figure 15: Relative location (% of tokens after the position) of the first top-1 difference or early KL divergence under steering; higher percentage indicates earlier effect of backward planning.

And for the position of first divergence metrics (percentage of tokens after first rank switch/high KL), differences between model sizes are much less pronounced than for other metrics.

Table 2: Vocabulary size and fraction of single token rhyming words. The fraction computed on the basis of unique words from each rhyme family both in the training and the test examples. Rhyming words that repeat across examples are counted only once.

| MODEL FAMILY | $|V|$ | % SINGLE TOKEN |
|---|---|---|
| GEMMA2 | 256K | 0.72 |
| GEMMA3 | 262K | 0.67 |
| LLAMA3 | 128K | 0.51 |
| QWEN3 | 152K | 0.51 |

## G TOKEN PROCESSING

Our findings and their interpretation could depend on the way the input is processed when it is tokenized and embedded. In this appendix, we report information relevant for the processing of rhyming words in terms of tokens.

First of all, different model families use different tokenizers. Indeed, Gemma models use tokenizers with larger vocabularies, which might help them learn richer token representations. Those in turn might contribute to better rhyming capabilities. However, our main findings regarding the influence of model size and instruction-tuning are not affected by differences in tokenization, because all models of the same family use the same tokenizer regardless of instruction-tuning and size.

We interpret our approach to steering as replacing the features of the planned rhyme family. However, if all words in a rhyme family shared the last token, we could have been effectively replacing the embedding of the last token itself as opposed to (presumably) rhyming features. In reality, most of the rhyming words are not split into multiple tokens, see statistics in Table 2. Furthermore, even when a rhyming word is split into more than one token, the last token is not always shared across words. For example, Gemma2 tokenizer treats most words that rhyme with *rare* as single tokens, except for *fla-ir*, *g-lare*, and *des-pair*; only in the last case does the last token coincide with another word from the same rhyme family.

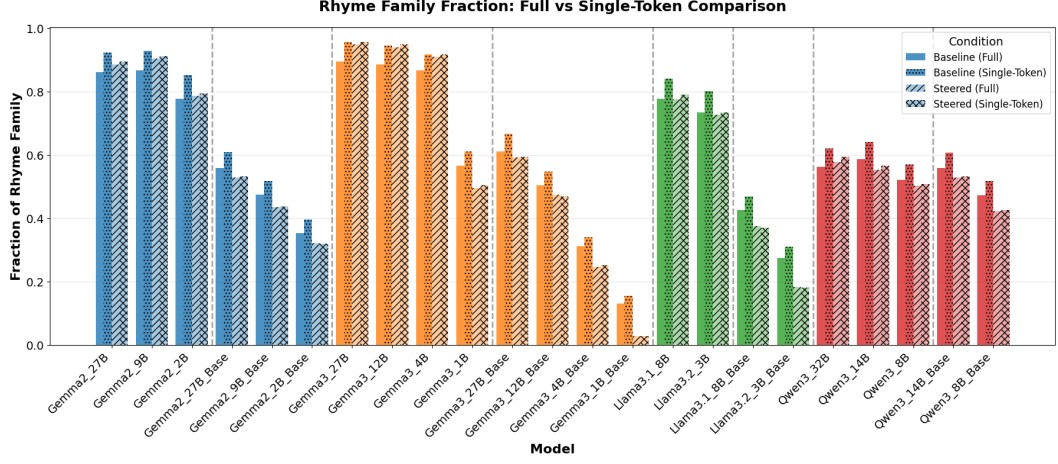

Figure 16: Baseline rhyming abilities of models vs. steered rhyming behavior. For each model, the bars report, respectively: 1. Frequency of rhyme family 1 (basic rhyming behavior), full test data. 2. Frequency of rhyme family 1 (basic rhyming behavior) for test examples where the last word of the first line is a single token in all models. 3. Frequency of rhyme family 2 under steering (steering effectiveness), full test data. 4. Steering effectiveness for test examples where the last word of the first line is a single token in all models.

To further control whether differences in tokenization significantly affect our results, we report our key metrics for the subset of test data where the last word of the first line in a couplet is always

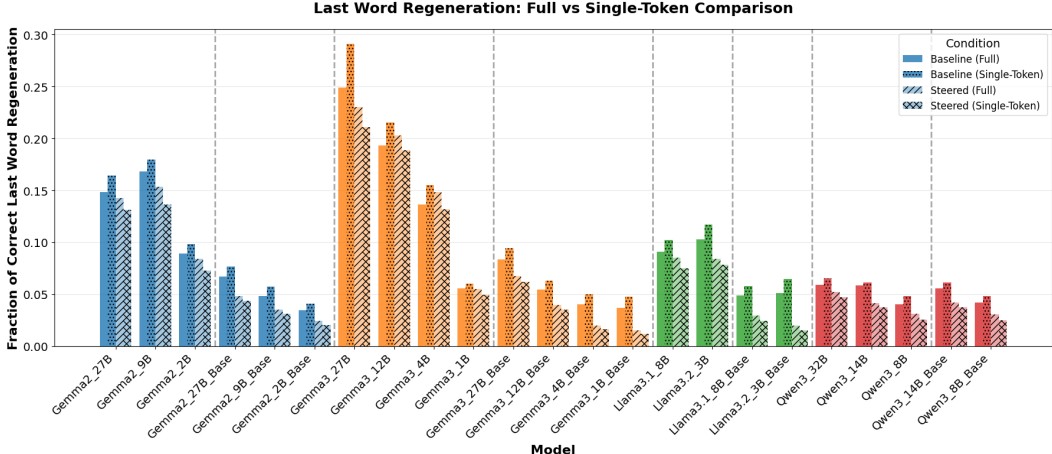

Figure 17: Last word regeneration rate of different models. For each model, the bars report, respectively: 1. Frequency of rhyme family 1 in regeneration, full test data. 2. Frequency of rhyme family 1 in regeneration for test examples where the last word of the first line is a single token in all models. 3. Frequency of rhyme family 2 in regeneration from lines produced under steering, full test data. 4. Frequency of rhyme family 2 in regeneration from lines produced under steering for test examples where the last word of the first line is a single token in all models

tokenized as a single token across all models. As one can see in figures 16 and 17, this filtering does not affect the results substantially. Some small consistent changes may be attributable to frequency effects. For example, single token words are a little easier to rhyme, which could be related to the fact that single token words are usually more frequent.

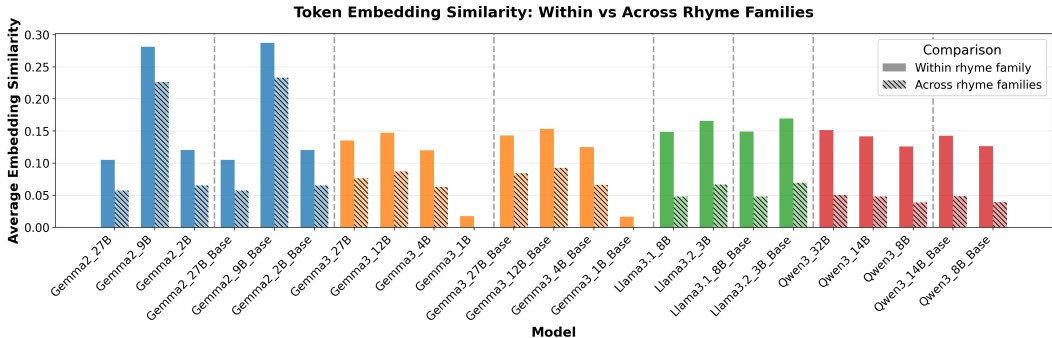

Figure 18: Embedding similarities of the last token of rhyming words across model. The plot shows average cosine similarity within rhyme family, compared to baseline similarities of last tokens across rhyme families.

Finally, differences in rhyming behavior could theoretically be attributable to the prominence of phonetic features in token embeddings of different models. We control for this by comparing average token embedding similarities for the last tokens of words within the same rhyme family vs across rhyme families. This comparison, reported in Figure 18, suggests that token embedding structure does not play a major in the differences across models: cosines are similar for base and instruction tuned models, as well as across model families and sizes. For Gemma2 9B, the cosine similarities within rhyme families are higher, but so are similarities across rhyme families. The only outliers are Gemma3 1B models where token embeddings within rhyme families are relatively dissimilar. This might contribute to Gemma3 1B's relatively low rhyming metric values; note however that instruction-tuned Gemma3 1B still rhymes better than base LLaMA and Qwen models, and slightly better than instruction-tuned Qwen3 8B.

## H    SPECIFIC WORD STEERING IN POETRY

We did multiple experiments trying out single word steering: steering the model to say a specific word at the end of the second line, such as *rabbit* or *habit* in Lindsey et al. example. We document one such experiment here, in which we steer Gemma2 9B and Gemma3 27B to end the second line with either the word *night* or the word *light*. To do this, we use Claude 3.5 Sonnet to generate 20 couplets ending in *night* and 20 couplets ending in *light*. We prompted Claude to write the first line, such that it is very suggestive for the word specific word. We then generated second lines for each prompt 500 times to estimate the probability that it would end in the correct word which we call suggestibility. We filtered out all prompts with a suggestibility below 0.8, which left a handful prompts on both sides. We used those to calculate the steering vectors on the newline token as described in the paper. We steered on the following prompt:
`\A rhymed couplet:\nThe forest path seemed to shrink quite tight\n"`
estimating the probability of the second line ending in either *light* or *night* (500 samples).

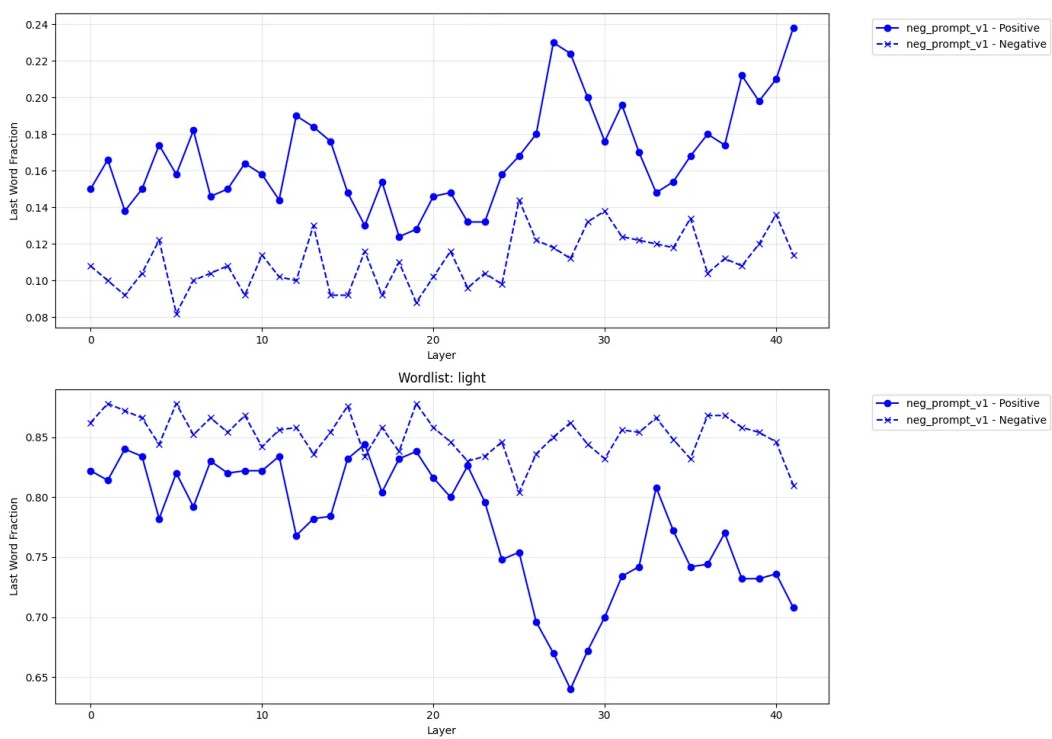

Figure 19: Fraction of second line ending in Light/Night when steered (Gemma2 9B)

Steering could only change the probability of a certain word by at most  20 percent for Gemma2 9B and  50 percent for Gemma3 27B, lower effect than what we can get in rhyme family steering. This and similiar experiments suggest that specific word forward planning may emerge with size of the model; single word steering works very little for the models we analyze but might work better for larger models like Claude Haiku.

## I    OTHER STEERING METHODS

While we opted for the simple average activation difference steering, it is also possible to use other types of vectors for steering, relying on embedding components encoding phonetic information. There is indeed support for phonetic features encoded systematically in LLM's embedding space McLaughlin & Merullo (2025).

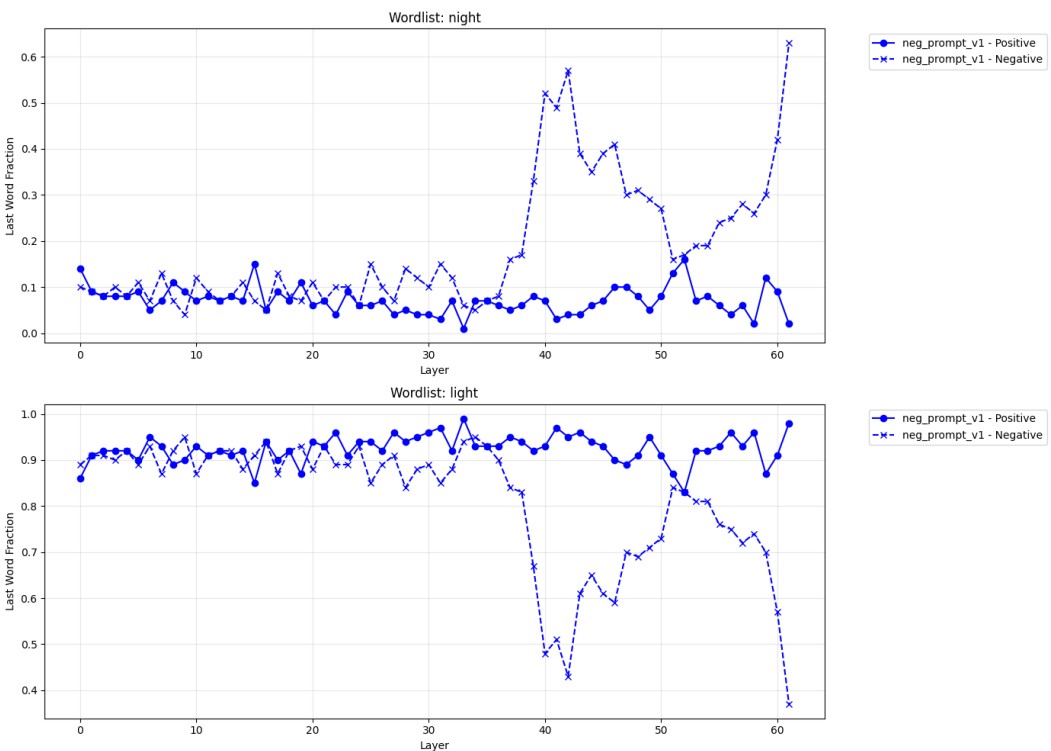

Figure 20: Fraction of second line ending in Light/Night when steered (Gemma3 27B)

One approach is to train a probe on residual stream activations for classifying rhyme families. Probe weights for a specific rhyme family act as its representation. Steering can then proceed using the difference of class weights of the target and source rhyme families.

It is also possible to identify sparse autoencoder latent dimensions that correspond to rhyme families and use them for steering. For instance, we found that latent 5862 in GemmaScope 16K SAE for residual stream of layer 20 in Gemma2 9B tends to fire on words that rhyme with *night*, and that latent 14069 fires on words that rhyme with *deep*. The difference between the decoder weights of the two can be used to steer generation between the rhyming families.

Both probing and SAE based approaches have the benefit of producing representations for specific rhyme families and as such open paths for further representation analysis. However, both of these approaches come with additional complexity, in particular requiring tuning the steering coefficient in each case. For this reason, we chose the simple and robust mean activation difference approach for our experiments, which allows to easily scale experiments across models, layers, and position.

Some examples of steered generation with alternative approaches (probe weights and SAE latents):

Prompt:

```
"A rhymed couplet:\nThe house was built with sturdy, reddish brick\n"
```

Baseline outputs, Gemma2 9B instruction tuned:

```
 "A rhymed couplet:\nThe house was built with sturdy, reddish brick\n
 And stood for ages, strong against the trick.\n\n\n
 Let me know if you'd like another couplet or a different rhyming scheme!\n",
'A rhymed couplet:\nThe house was built with sturdy, reddish brick\n
And stood for many years, a steadfast trick.\n\n
**Explanation:**\n\n* **Rhyme:** "brick" and "trick" rhyme, creating',
'A rhymed couplet:\nThe house was built with sturdy, reddish brick\n
```

```
And stood for years, a testament to stick.\n\n
This is a couplet because it consists of two lines of poetry that rhyme. \nIt',
"A rhymed couplet:\nThe house was built with sturdy, reddish brick\n
And stood for years, impervious to trick. \n\n\n
Let me know if you'd like more! \n\n"
```

Outputs with steering on newline token with difference of probe weights for -ick rhymes and -ore
rhymes, steering on layer 27, steering multiplier 400:

```
'A rhymed couplet:\n
The house was built with sturdy, reddish brick\n
A testament to love, a timeless, sturdy core. \n\n\n
Please provide another rhyming couplet that continues the theme of a home built w
"A rhymed couplet:\n
The house was built with sturdy, reddish brick\n
And stood steadfast, defying winter's core.\n\n
Let me know if you'd like to explore more poems!\n",
'A rhymed couplet:\n
The house was built with sturdy, reddish brick\n
And stood for years, a testament to yore.\n\n
**Explanation:**\n\n
* **Rhyme:** The words "brick" and "fore" rhyme',
'A rhymed couplet:\n
The house was built with sturdy, reddish brick\n
And stood through storms, a testament to lore.\n\n
This is an example of a couplet, a two-line stanza in poetry with a rhyme'
```

Prompt:

```
"A rhymed couplet:\nIn silence, the strongest hearts sometimes softly weep\n"
```

Baseline outputs:

```
"A rhymed couplet:\n
In silence, the strongest hearts sometimes softly weep\n
For unspoken sorrows, secrets they gently keep. \n\n\n
Let me know if you'd like more! \n\n",
"A rhymed couplet:\n
In silence, the strongest hearts sometimes softly weep\n
While the bravest souls their hidden sorrows keep. \n\n\n
Let me know if you'd like to explore more poetry!\n",
"A rhymed couplet:\n
In silence, the strongest hearts sometimes softly weep\n
For a love that's lost, a secret they keep.\n\n\n
Let me know if you'd like more couplets!\n",
'A rhymed couplet:\n
In silence, the strongest hearts sometimes softly weep\n
While smiles mask the tears that in secret they keep.\n\n\n\n',
"A rhymed couplet:\n
In silence, the strongest hearts sometimes softly weep\n
And hidden tears, like pearls, their secrets keep.\n\n\n
Let me know if you'd like more poetic explorations!\n"
```

Steering on the newline token on layer 20, steering multiplier 100 produces the target rhyme in 832
out of 1000 rollouts when steering with difference of SAE decoder weights for latents 14069 (*-eep*)
and 5862 (*-ight*) from GemmaScope 16K. Output examples:

```
"A rhymed couplet:\n
In silence, the strongest hearts sometimes softly weep\n
For dreams that vanished in the lonely, darkest night. \n\n\n
```

```
Let me know if you'd like more!\n",
'A rhymed couplet:\n
In silence, the strongest hearts sometimes softly weep\n
Tears, unshed, held back by a will, strong and bright. \n\n\n',
'A rhymed couplet:\n
In silence, the strongest hearts sometimes softly weep\n
When hidden burdens weigh heavy, day and night. \n\n\n\n',
"A rhymed couplet:\n
In silence, the strongest hearts sometimes softly weep\n
For burdens unseen, hidden from day's bright light. \n\n\n
Let me know if you'd like more!\n\n",
"A rhymed couplet:\n
In silence, the strongest hearts sometimes softly weep\n
For losses unseen, yet felt with all their might.\n\n\n
Let me know if you'd like more!
I can write you more couplets on different topics.\n"
```

## J SAE LATENTS AND RHYME PLANNING

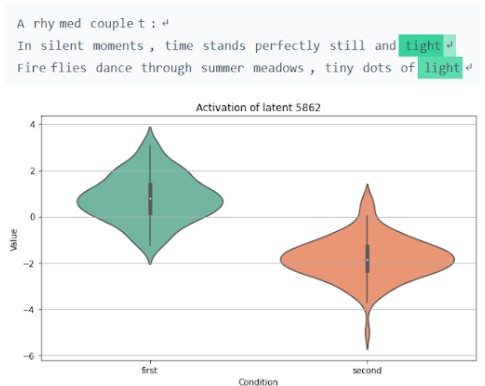
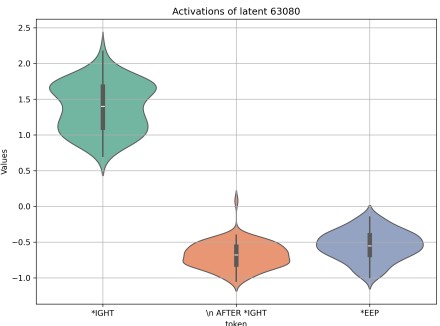

Figure 21: **Top left**: SAE latent 5862 (layer 20, GemmaScope 16K) fires on words from the *-ight* rhyme family, but also on the newline token at the end of the first line of a couplet where rhyme for the next line can be planned. **Bottom left**: Firing on the first but not the second newline is systematic, as shown on a sample of 100 couplets with *-ight* rhymes. **Right**: In contrast to Gemma2 9B's 5862, latent 63080 in Llama 3.1 8B fires on words of the *ight* rhyme family, but its activations on the following newline are comparable to its activations on words from a different rhyme family.

Some suggestive observations on sparse autoencoder latents provide further evidence for planning in rhyme generation.

As mentioned above, latent 5862 (GemmaScope 16K layer 20) corresponds to words that rhyme with *night*, and we can use it in steering to produce that rhyme. We observe further that latent 5862 tends to fire both on the last word of the first line of a couplet (e.g. *light*) and on the following newline token. These are the two positions that support rhyme family steering in Gemma2 9B, thus involved in the rhyme planning circuit. Latent 5862 is not activated on the newline token after the second line of a couplet, where rhyme planning is not needed.

We identify for Llama 3.1 8B, layer 25, latent 63080 with a similar function (fires on words rhyming with *night*). however, latent 63080, unlike its Gemma2 9B counterpart, is not activated on newline tokens. This is consistent with the fact that Llama does not support rhyme family steering on the newline token. See Fig. 21 for an illustration.

## K NOUN QUESTION ANSWER STEERING

We further investigated the steering for an answer across the full range of 23 language models.

We created a dataset of questions that allows us to detect whether the manipulating the planned answer to a question using our steering methodology affects the generated answer as well as the choice of the article form. For example, we apply a targeted intervention to LLM activations in a question whose answer is "an eye". The steering is applied locally to a position at which the answer is not yet generated, such as the last word of the question. If we can steer the model to answer "heart", this presents evidence for forward planning. If we further find that the model chooses article form *a* rather than *an* before generating *heart*, this constitutes evidence for forward planning.

Our dataset includes of 20 common nouns, of which 10 begin with a vowel and 10 begin with a consonant. For each noun, we use 13 suggestive questions for estimating the steering vectors and 5 suggestive questions for testing the steering. We also created a second test set of 7 questions per noun pair that are meant to be (more) neutral.

Steering vectors for noun pairs were estimated via mean activation difference for the 13 questions from the train set for each noun. The best steering location was selected among the 80% middle layers of each model and three positions: question mark token, the position after it (newline token), and the position before it (last word). The best layer-position combination was used for the results reported.

All models showed the best steering on one of the middle layers. The best steering location in most cases was the last word. For LLaMA and Qwen models in neutral questions and for Qwen3 14B Instruct in word-suggestive questions the best steering position was the question mark. See figures 31-34 for details on steering effect across layers and positions.

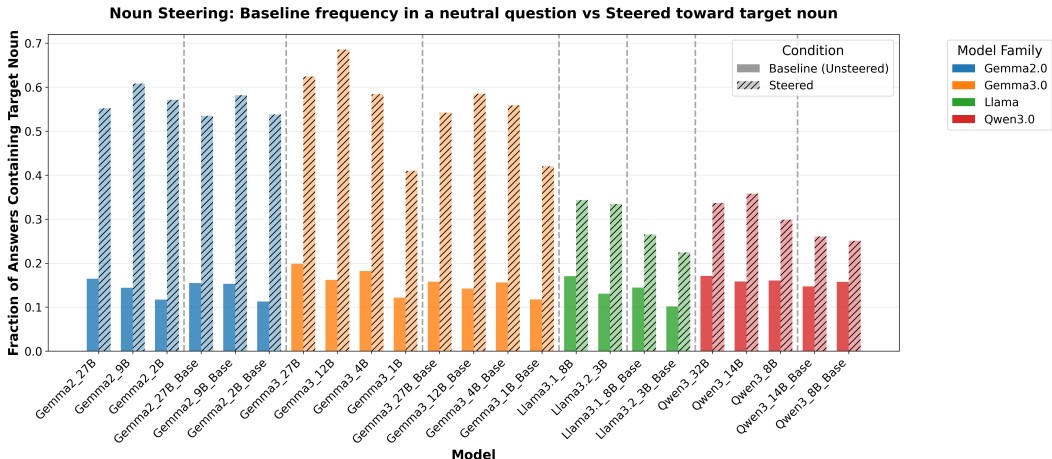

Figure 22: Percentage of generated answers containing the noun in baseline answers to neutral questions vs. steered towards the target noun. In this setting, steering recovers only a fraction of the target noun's baseline frequency.

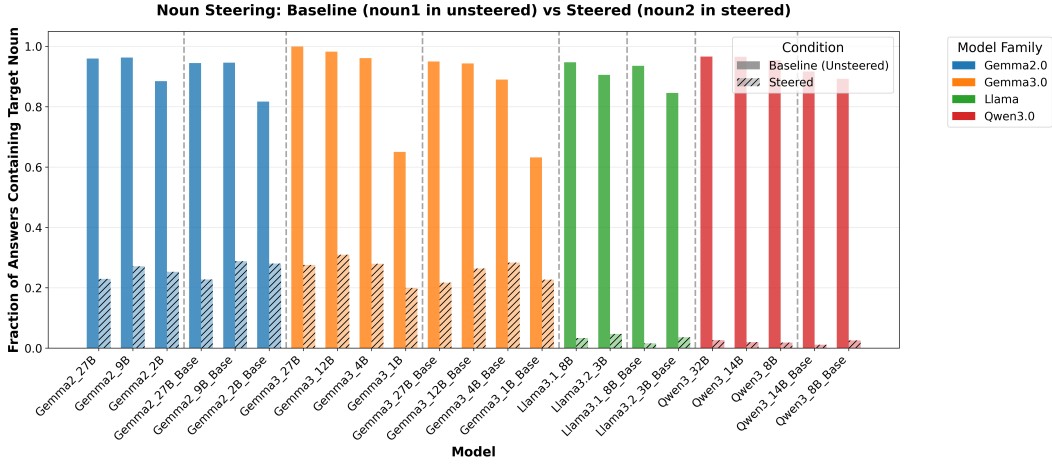

Figure 23: Percentage of generated answers containing the noun in questions for that noun vs. steered towards the target noun in questions for the other noun in the pair. In this setting, steering recovers only a fraction of the target noun's baseline frequency.

Steering with mean activation differences of questions for a pair of nouns reliably increased the frequency in the answer of the target noun steered towards, see figures 23 and 22 (note that the questions themselves do not contain the target nouns, but are meant to elicit the nouns as answers). Steering was far from fully recovering the baseline noun frequency for suggestive questions, but worked quite well in affecting the answer in neutral questions. This suggests that while answer planning is probably distributed across positions and layers so a single position-layer is not enough to capture it, we did intervene on an important location, manipulating **forward planning** for the answer. Like in the rhyming study, instruction-tuned models show a stronger steering effect than base models in neutral questions, suggesting that instruction tuning boosts planning. For suggestive questions, patterns are much less clear, presumably because of planning is more distributed across positions in informative contexts.

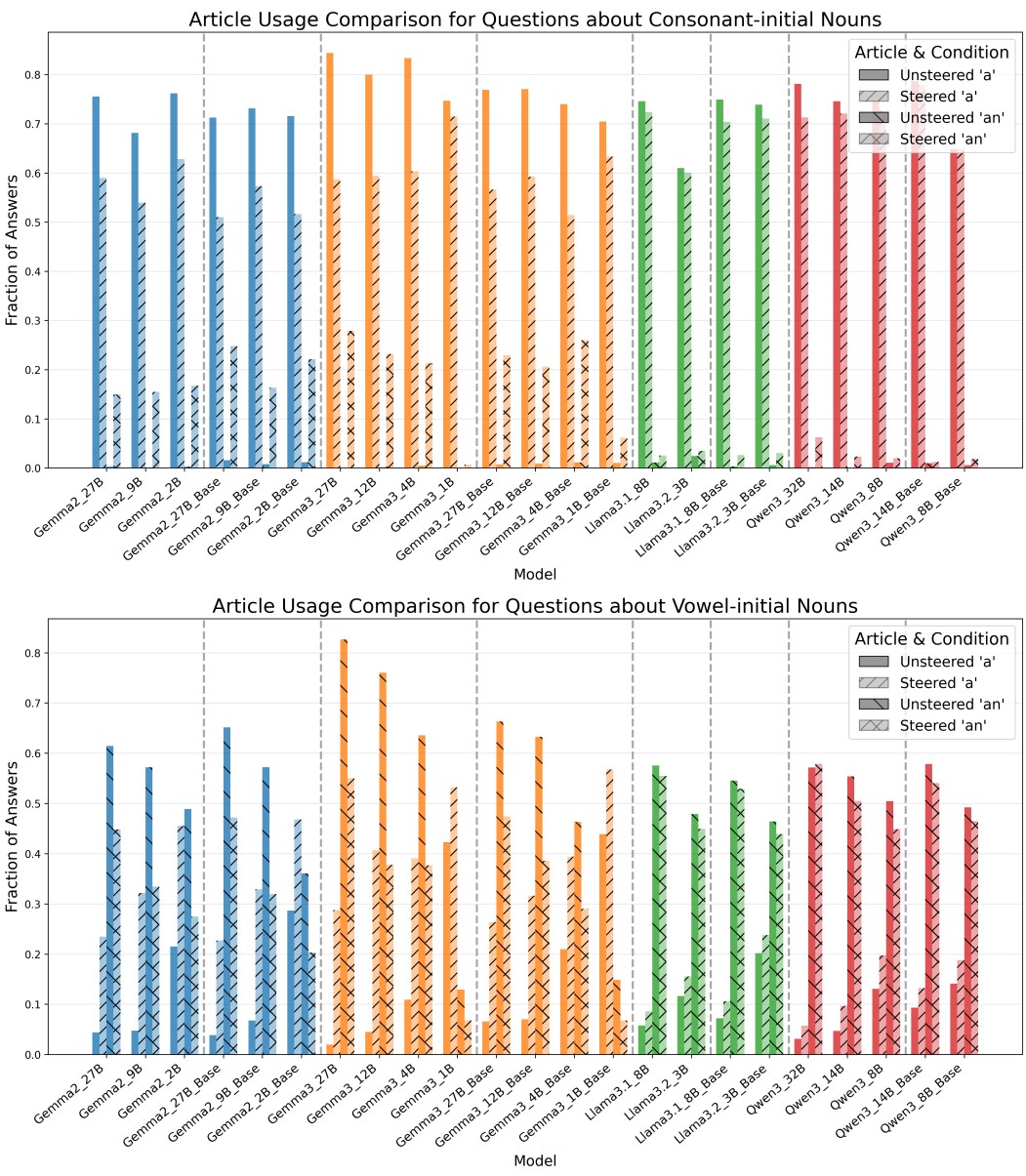

Figure 24: Percentage of answers containing article *a* or *an* as a function of steering on **suggestive questions**. Proportion of *an* increases and proportion of *a* decreases when steering towards a noun beginning with a vowel in a question about a noun beginning with a consonant, and vice versa when steering towards a noun beginning with a consonant in a question about a noun beginning with a vowel.

The word pairs were specifically selected to contain one answer noun that begins with a consonant (and goes with article *a*) and one noun that begins with a vowel (and goes with article *an*). We measure if articles used vary with the target noun when steering. Indeed, the frequency of article forms is modulated predictably by steering both in suggestive questions (fig. 24) and neutral questions (fig. 25), providing evidence for **backward planning** in question answering context.

Figure 25: Percentage of generated answers to neutral questions containing article *a* vs. *an* as a result of steering on **neutral questions**. Lines connect data points for steering conditions of the same language model.

## L    MORE COMPLEX QUESTION ANSWERING AND SUBJECT-VERB AGREEMENT AS A TEST OF IMPLICIT PLANNING

Similar to Hanna & Ameisen (2026), we constructed a dataset to investigate implicit planning in the context of subject-verb agreement:

> *Example* 3.  Question: There were 7 keys but 4 were lost.  How many keys are left?\n\n Answer: Now there

All dataset examples follow a similar pattern to Example 3, varying the nouns and corresponding verbs. LLM's typically produce a verb form of "to be" followed by a number as the next tokens—in this case, "are 3". This task appears to require implicit backward planning: the model must generate the correct intermediate token (either "is" or "are") before producing the goal token (the correct number).

The model could solve this task through in two distinct ways. The first is **improvisation**, where some circuits in the model determine the correct verb form without representing the planned answer. The second is **planning**, where the model has some intermediate representation of the correct answer and uses this representation to generate both the appropriate verb and the answer itself. Our experiments provide suggestive evidence for the planning mechanism.

We evaluated this task on Gemma3 27B using a dataset of 80 questions: 40 with answer "1" and 40 with answer "3", split into 30 training examples (for estimating steering vectors) and 10 test examples.

Table 3 demonstrates that Gemma3 27B reliably produces the correct verb form in the baseline condition. Crucially, steering on the question mark token can shift the model to produce the steered verb form in a substantial fraction of cases. Additionally, steering occasionally succeeds in changing

| Metric | Answer is 1 | Answer is 3 |
|---|---|---|
| Correct Verb (%) | 97.5 | 100.0 |
| Correct Number (%) | 98.4 | 100.0 |
| Steered towards Verb (%) | 52.8 | 9.4 |
| Steered towards Number (%) | 0.0 | 9.0 |

Table 3: Accuracy and steering effectiveness of Gemma3 27B on the subject-verb agreement task. Baseline metrics show the model's ability to produce correct verbs and numbers. Steering metrics show the percentage of cases where steering on the question mark token successfully shifts outputs toward the target verb form and corresponding number.

the models answer from "3" to "1". For some prompt this effect is quite large. For The example prompt in 3, steering causes the model to output "is" in 93% of cases and "1" in 58% of cases.

If the improvisation mechanism were true, then we would expect our steering to only be able to manipulate the verb and the effect on the answer token coming entirely though the indirect effect of changing the word. To test this, we conduct another small experiment where we directly append "is" to the input prompt instead of using activation steering. In this case, the model never outputs "1". This suggests that at least in some cases, our steering vector captures a representation of the planned answer ("1"), which the model then also uses to generate the correct verb form, providing evidence for implicit planning rather than mere improvisation.

## M  STEERING EFFECTIVENESS ACROSS LAYERS AND POSITIONS

In this appendix, we include plots with more detailed information on the positions and layers that are most effective is our steering experiments.

### M.1  RHYME STEERING

Figure 26 compares the effectiveness of steering on the newline token for all models. Figures 27-30 report steering effectiveness for both the newline position and the immediately preceding token position for all models in consideration.

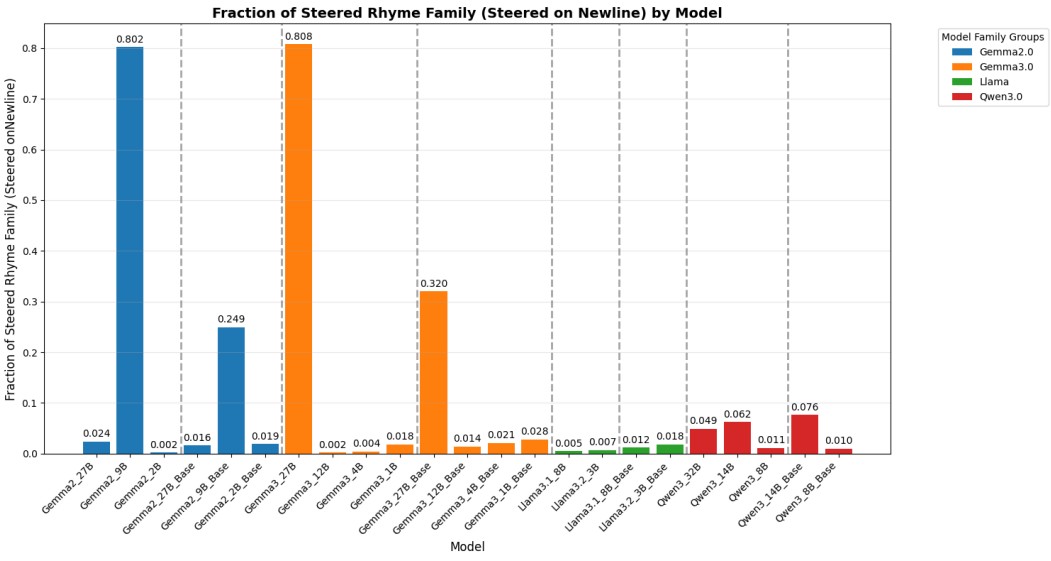

Figure 26: Steering effectiveness for newline token position, all models.

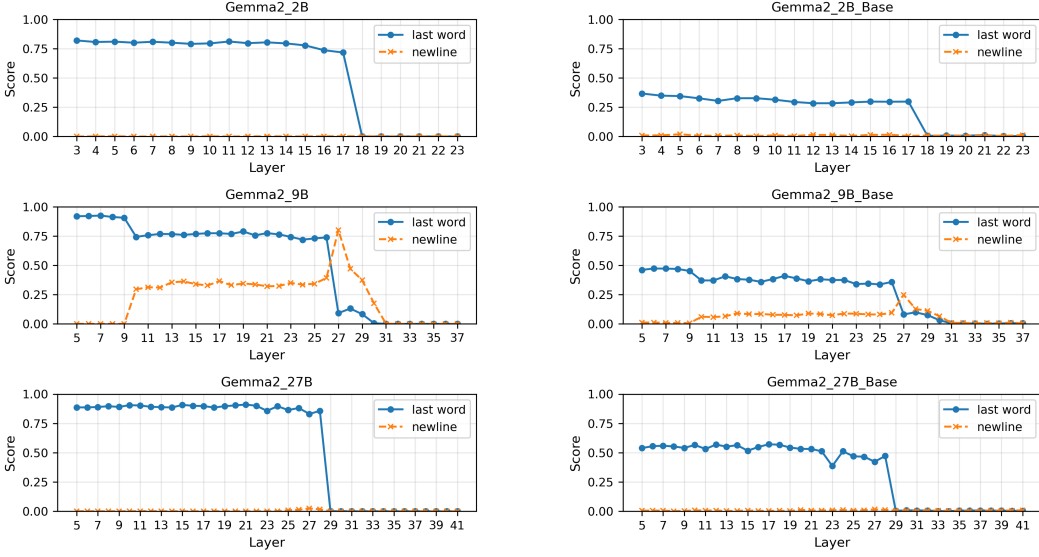

Figure 27: Steering effectiveness by steering position and layer, Gemma2 models.

Lindsey et al. reported evidence for planning representations on the newline token at the end of the first line for Claude Haiku. In our experiments, steering on the newline token was only effective in

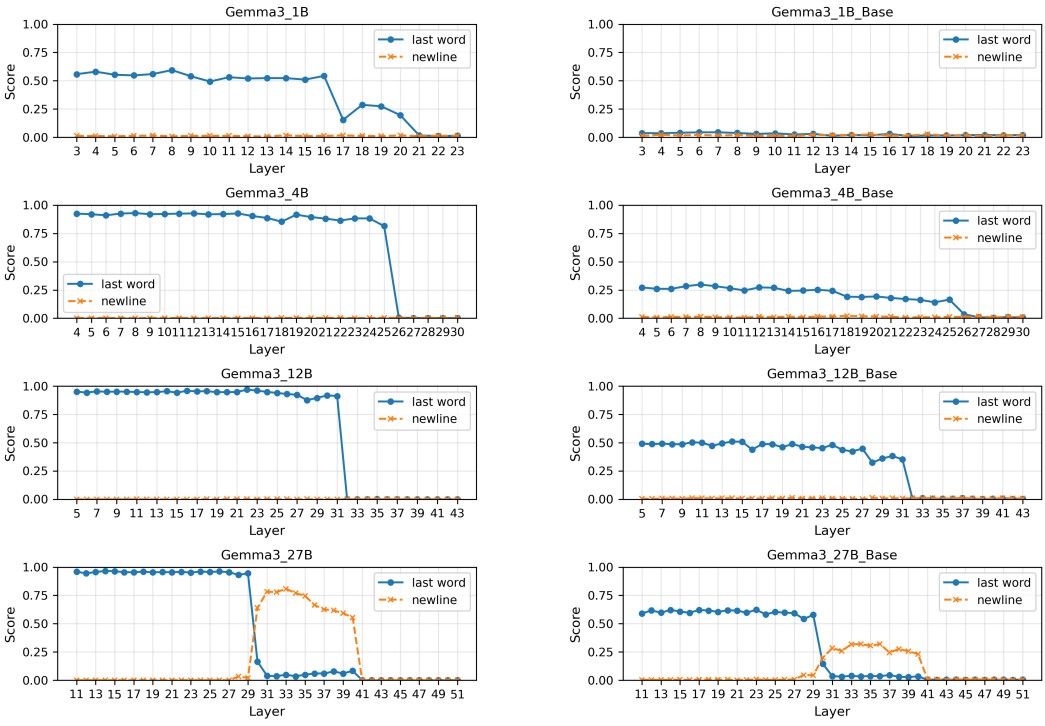

Figure 28: Steering effectiveness by steering position and layer, Gemma3 models.

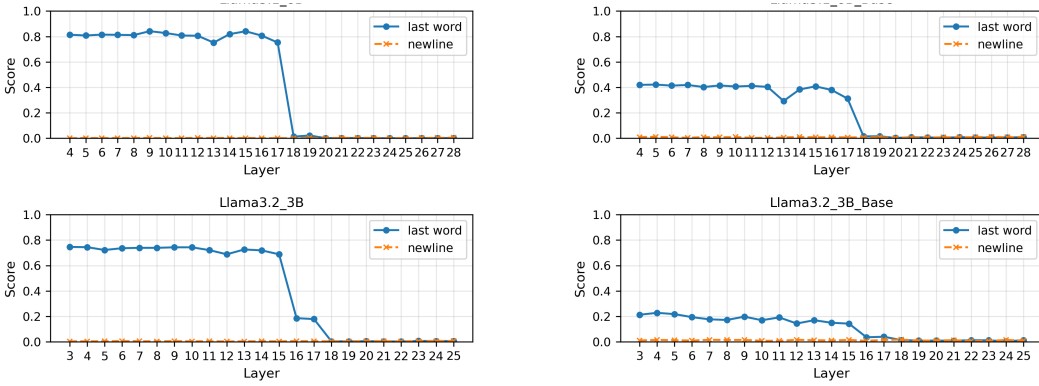

Figure 29: Steering effectiveness by steering position and layer, Llama3 models.

select models in some middle layers. For all models however, steering worked on the pre-newline (last word) token in earlier layers. Rhyme steering effectiveness for all layers and positions of all models is reported in figures 27–30.

## M.2 NOUN QUESTION ANSWERING STEERING

Noun QA steering effectiveness for all layers and positions of all models is reported in figures 31–34.

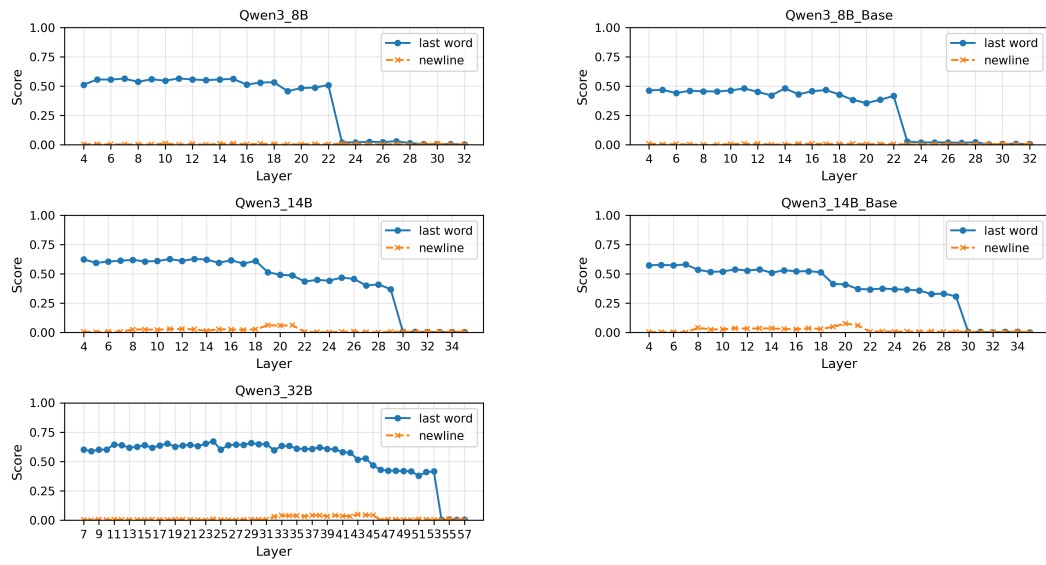

Figure 30: Steering effectiveness by steering position and layer, Qwen models.

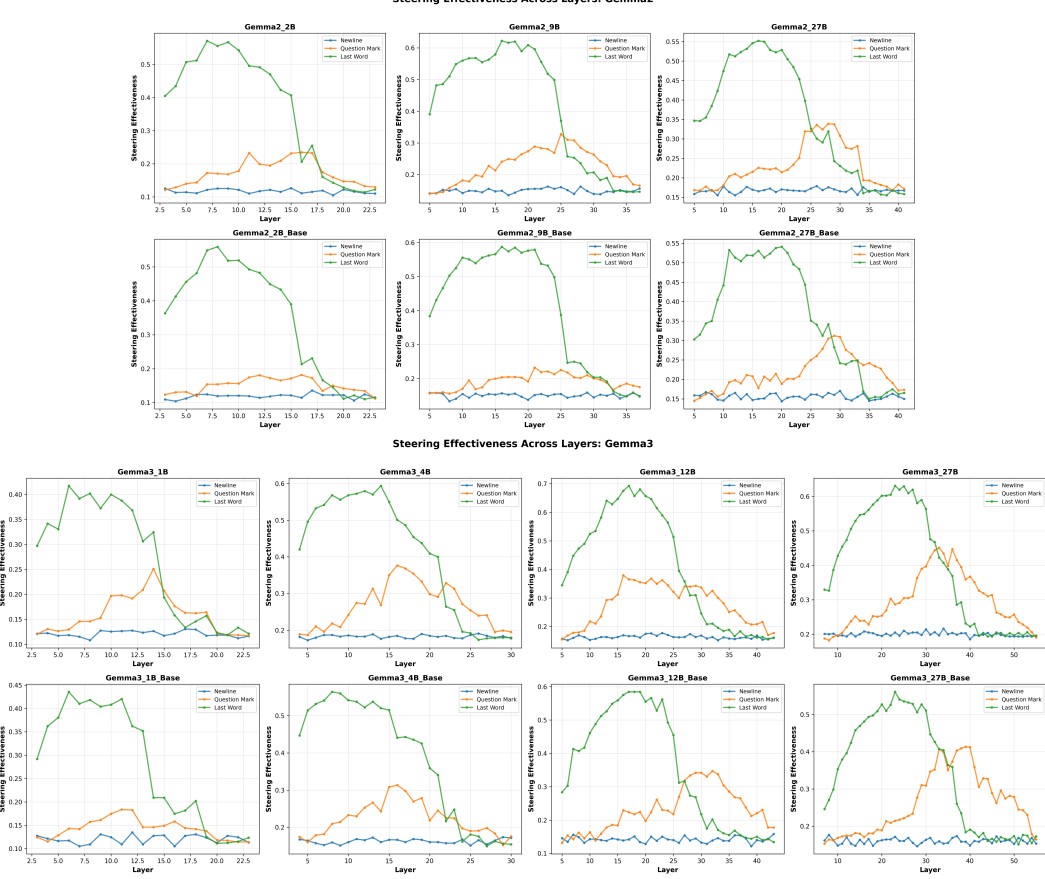

Figure 31: Steering effect (fraction of rollouts containing the target noun) across layers and positions for steering. Neutral questions, Gemma models.

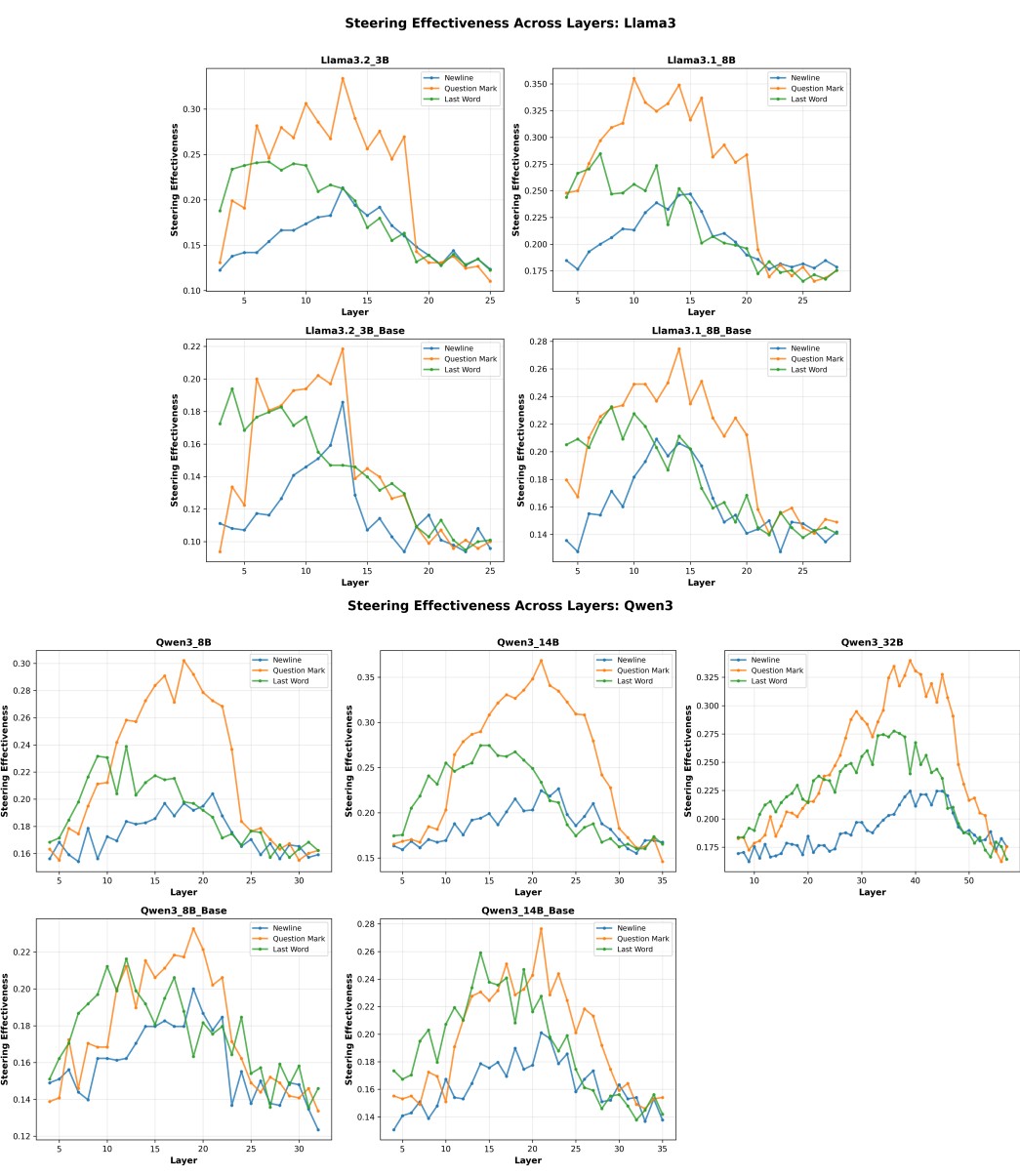

Figure 32: Steering effect (fraction of rollouts containing the target noun) across layers and positions for steering. Neutral questions, Llama and Qwen models.

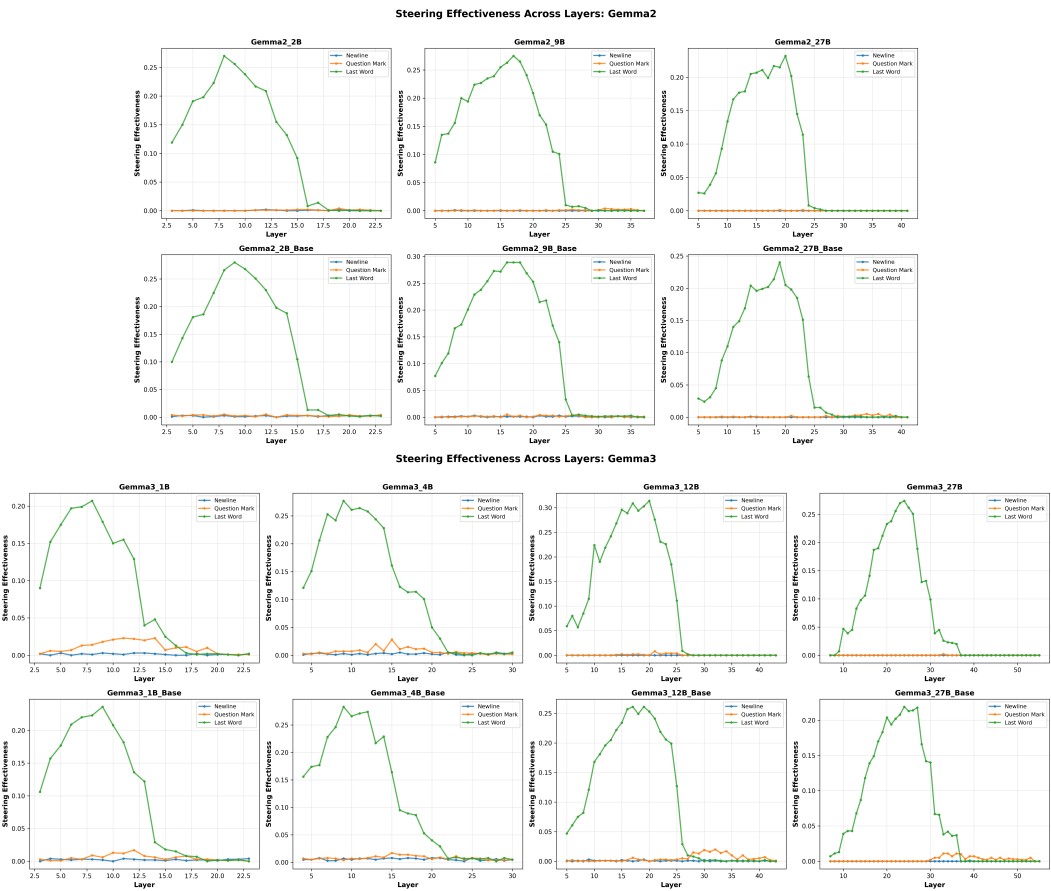

Figure 33: Steering effect (fraction of rollouts containing the target noun) across layers and positions for steering. Suggestive questions, Gemma models.

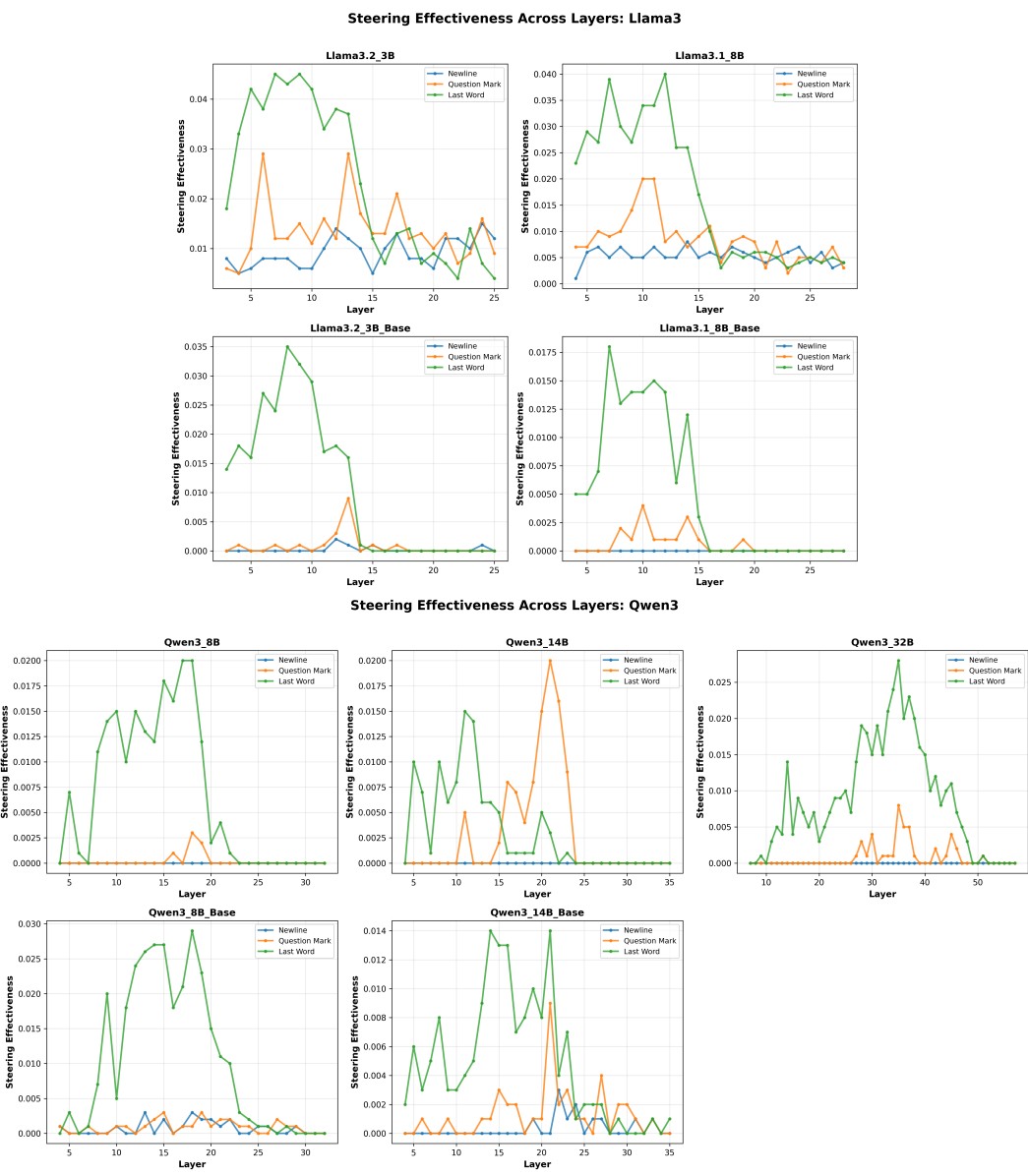

Figure 34: Steering effect (fraction of rollouts containing the target noun) across layers and positions for steering. Suggestive questions, Llama and Qwen models.

