# OpenReview forum: "What's the plan? Metrics for implicit planning in LLMs and their application to rhyme generation and question answering"
_ICLR.cc/2026/Conference — ICLR 2026 Poster_

### Official Review · Reviewer_rPUj · 2025-10-23

**Soundness:** 2
**Presentation:** 2
**Contribution:** 2
**Rating:** 4
**Confidence:** 3

**Summary:**

This paper investigates whether LLMs exhibit implicit planning, meaning that they internally represent and use information about future outputs during text generation. Using rhyme generation as a controlled testbed, the authors introduce formal definitions and quantitative metrics of forward planning (encoding future goals before generation) and backward planning (conditioning ongoing generation on future goals). By manipulating hidden activations with mean activation steering, they show that altering representations at specific positions (e.g., line endings or newline tokens) systematically changes future rhyme families without modifying the input text, demonstrating controllable planing vectors. Experiments across various language models show that implicit planning behaviors are consistently observable and become more pronounced in larger and instruction-tuned models. The study further provides evidence that such planning mechanisms are structured and interpretable within model internals.

**Strengths:**

The authors' proposed steering vector based on the average activation difference appears to be directly effective, reflecting to some extent the underlying correlation between model activations and rhyme families.

**Weaknesses:**

1. The paper confines all of its experiments to the rhyme generation task, a setting that *has already been widely examined* in prior research. This narrow focus limits the generalizability of the findings and does not provide sufficient evidence to confirm the existence of an implicit planning mechanism.
2. The distinction between forward planning (the direct influence on the final output) and backward planning (indirect influence through the evolution of token probabilities) appears to describe two stages of the same representational process rather than two independent behaviors. The *lack of discussion on the connection* between these two aspects weakens the conceptual clarity of the work.
3. The proposed method and evaluation metrics are *relatively primitive and overly direct*. Modifying activation values essentially alters the contextual representation itself, making it difficult to isolate the planning effect. The experiments would benefit from deeper analyses, including an examination of cross-layer activation propagation and the relationship between attention distributions and rhyme outcomes.
4. The overall organization of the paper is confusing, and the figures are roughly presented and difficult to interpret. These presentation issues significantly reduce the clarity and accessibility of the work.

[1] Jack Lindsey, etc,. On the biology of a large language model. Transformer Circuits Thread, 2025.
[2] He, Jingkai, et al. History rhymes: Accelerating llm reinforcement learning with rhymerl. arXiv preprint arXiv:2508.18588, 2025.
[3] Jobanputra, Vedant Dhaval, et al. LLM-aided Evolutionary Algorithms for Haiku Generation. Proceedings of the Genetic and Evolutionary Computation Conference Companion. 2025.

**Questions:**

What is the relationship between the forward planning and backward planning proposed in this paper?

---

> ### Author Response · Authors · 2025-11-14
>
> Dear Reviewer rPUj,
>
> We thank you for a thoughtful feedback on our initial submission. Before we are able to reply more substantively, we would like to ask you to clarify your suggestion under (3): "The experiments would benefit from deeper analyses, including an examination of cross-layer activation propagation and the relationship between attention distributions and rhyme outcomes". One of us interprets this as a suggestion to add an analysis via attention ablation at specific layers in the absence of steering, blocking information flow from the end of the previous line, and detecting its effect on output token prediction in the second line (intermediate positions and the end of the line). Is this interpretation consistent with what you intended, or did you have something else in mind?
>
> Thank you in advance for your response,
> Authors of Submission21127

---

> > ### Comment · Reviewer_rPUj · 2025-11-28
> >
> > Sorry for the late reponse. Yes, you undetstand it correctly.

---

### Official Review · Reviewer_ArCv · 2025-10-28

**Soundness:** 4
**Presentation:** 4
**Contribution:** 4
**Rating:** 8
**Confidence:** 5

**Summary:**

This paper investigates the concept of "implicit planning" in LLMs through the specific case study of rhyming poetry generation. The authors propose a set of simple, scalable metrics to quantify both forward planning (preparing for a future rhyme) and backward planning (adjusting intermediate tokens to lead to that rhyme). Using mean activation difference steering, they demonstrate the ability to manipulate the "planned" rhyme family across a diverse set of 23 open-weight models. The results show that planning capabilities for rhyming are present even in 1B parameter models and generally improve with model scale and instruction tuning. The paper concludes with a brief mechanistic analysis of the "rhyming circuit" in Gemma2 9B and with identifying specific attention heads involved in the process.

**Strengths:**

The paper's core strength is its departure from computationally intensive methods like "cross-layer transcoders" (I was not aware of these until this paper). The use of mean activation steering and straightforward metrics makes this work easily replicable and applicable across a vast array of models, as demonstrated by the authors' evaluation of 23 different models. There's a rich amount of steering literature to base this all on.

The study isn't limited to a single large model but instead provides a panoramic view of how rhyming and planning abilities evolve across different model families (Gemma, Llama, Qwen) and sizes (1B to 32B). This comparison between and discussion around base and instruction-tuned models is particularly insightful.

The authors don't rely on a single metric. The combination of generation-based metrics (Fraction of Correct Rhyme), probability-based metrics (KL divergence, Top-1 difference), and regeneration metrics provides a multi-faceted and convincing case for both forward and backward planning - and "planning" at the log probs level might be controversial

The analysis in Section 5.2, which pinpoints specific attention heads responsible for propagating the planning information, adds a layer of mechanistic credibility to the quantitative results and aligns with findings in related work.

**Weaknesses:**

The most significant weakness is the complete failure to discuss the role of tokenization. The model's ability to "rhyme" is profoundly influenced by its subword vocabulary!!!! For rhyme families like "-ight", words like "light", "night", and "right" likely share a common final token, making the task a simple exercise in token-level sequence completion rather than true phonological planning! The paper's entire analysis is conducted without ever acknowledging or controlling for this massive confounding variable. This oversight calls into question the interpretation of "planning" for certain rhyme families. I'm being very generous with my high scores in spite of this.

The literature review is troublingly incomplete, which seems to have led to the first weakness. The paper misses several key pieces of prior work. Most notably, Roush et al. (2022) in "Most Language Models Can be Poets Too" dedicated significant discussion to the very problem of tokenization and vocabulary size impacting poetic and rhyming ability. It is a foundational paper for this specific topic and its omission is a serious oversight. Additionally, other works on steering model behavior, like SMC Steering by Lew et al, could have been mentioned to provide a broader context for the intervention techniques used.

Related to the tokenization issue, the paper notes that larger models are better but doesn't explicitly discuss why. A key hypothesis, supported by Roush et al., is that larger models tend to have larger vocabularies (i.e vocab size in subseqeunt versions of GPT has grown well beyond its original 50K tokens), meaning more words (and rhyming suffixes) are represented by single tokens, simplifying the rhyming task. This connection is a critical piece of the puzzle that the authors fail to mention!

I give a generous score given that I believe this paper in its current form has a glaring lack of discussion around Tokenization, however, I'm not convinced that this lack of discussion would invalidate the results and I think adding such a discussion is basically trivial, so I give a high score now - but if the authors do not commit to sufficiently cover tokenization in some way in the final manuscript I would be liable to reduce my score to weak accept.

**Questions:**

Could the authors comment on the role of tokenization in their findings? For rhyme families where the rhyming component is a single, common token (e.g., "-ight"), is the model truly engaging in "planning" in the same way it would for families where words are tokenized differently (e.g., words ending in "-ake" like "make", "take", "shake")? Have the authors analyzed if the performance of their metrics correlates with the token-level similarity of words within a rhyme family?

The circuit analysis points to specific heads in Gemma2 9B. Given the universal presence of the planning phenomenon across models, do the authors hypothesize that these are "rhyming heads" that are functionally homologous across different model architectures, or is it more likely that each model implements this capability with different, idiosyncratic circuits?

---

### Official Review · Reviewer_Z2sB · 2025-10-29

**Soundness:** 3
**Presentation:** 2
**Contribution:** 3
**Rating:** 4
**Confidence:** 4

**Summary:**

This paper investigates the implicit planning capability in large language models (LLMs), using rhyming poetry generation as the object of study. The authors propose a set of quantitative, scalable metrics to evaluate various aspects of implicit planning, including successful forward planning and successful backward planning.

The paper introduces the "mean activation difference steering" (MADS) method, which intervenes in the model's hidden states during generation to change the generated rhyme family. The authors created a dataset of rhyming couplets covering 10 rhyme families and used it to conduct a comprehensive evaluation of 23 open-source models (Gemma 2, Gemma 3, Qwen 3, Llama 3.1/3.2) across different series and scales (1B to 32B parameters).

**Strengths:**

- Clear Motivation: The paper is sharply focused on studying the implicit planning capability of LLMs, providing clear research intent.

- Simple and Effective Method: The paper proposes the mean activation difference steering method, which is simple and effective, serving as a replacement for more complex and computationally expensive techniques used in previous research.

- Extensive Experimentation: The authors validate their method across a wide range of popular model families and various model scales, leading to comprehensive experimental results.

**Weaknesses:**

- Writing Needs Improvement:

    - The paper's description of the activation manipulation position is somewhat confusing. Lines 157–159 mention that the manipulation target is the position of the newline character itself, while the immediately following formula on line 169 describes manipulation at the position preceding the newline character. This is confusing for the reader. Although the text immediately mentions that they experimented with both cases, I recommend the authors optimize the logical flow and presentation of this section.

    - Regarding the formula on line 169, the paper writes $P_{\text{RF}_2}^{\text{Train}} - P_{\text{RF}_1}^{\text{Test}}$. However, if I understand the proposed method correctly, the latter term should also be from the training set, meaning it should be $P_{\text{RF}_2}^{\text{Train}} - P_{\text{RF}_1}^{\text{Train}}$.

    - The figures in the paper are difficult to read due to small font sizes. It is recommended to adjust the typography in the visualizations.

- Limited Generalizability: The paper's study of LLM planning behavior is concentrated on rhyming poetry generation, and the generated content length is relatively short. Only a very brief discussion of Q&A tasks is included in the appendix. This limits the generalizability of the paper’s proposed metrics.

**Questions:**

- The proposed method appears similar to Classifier-Free Guidance (CFG) commonly used in diffusion models, but the paper does not mention this connection. Could the authors provide some discussion and analysis regarding the similarities or differences?

- How many training samples are required to obtain a relatively stable steering vector? The paper uses 85 samples. What would the effect of the steering vector be with significantly fewer samples, for example, less than 10?

---

### Author Response · Authors · 2025-12-03
**General response to reviews and overview of revisions. Part II**

# Reviewer rPUj

**Regarding task generality**: we added a comprehensive question answering task in addition to the rhyming task, see above.

**Regarding the relationship between forward and backward planning**: We agree this connection needed clarification. We have:
- Restructured the introduction to clearly articulate how these represent complementary aspects of the planning process;
- Added an explicit discussion of their interdependence.

**Regarding methodology**: Reviewer rPUj raises questions about the directness and simplicity of our approach, while acknowledging that it “appears directly effective” and that our study “provides evidence that such planning mechanisms are structured and interpretable within model internals”. As a remedy to their concerns, reviewer rPUj specifically requested attention ablations as the additional analysis needed. We implemented exactly this experiment (Appendix D), which aligns with their stated criteria for strengthening the paper. We further emphasize that the simplicity of our approach is intentional and advantageous:
- Our streamlined methodology enables the first large-scale investigation of implicit planning across multiple model families;
- The directness of our approach provides clear evidence for planning mechanisms.

**Regarding structure and readability**: We implemented a number of improvements, see above.

**Literature suggestions**: Reviewer rPUj suggests three papers.
- We already discussed Lindsey et al. 2025 at length in the original submission.
- We added a comparison with Jobanputra et al’s approach at the end of section 2.
- He et al. 2025 must have been suggested by mistake. The paper has nothing to do with rhyming beyond the name of the proposed reinforcement learning approach (RhymeRL).

# Summary

**Our revised manuscript directly addresses every point raised by the reviewers, which substantially strengthens its contributions**. Two of the reviewers expressed enthusiasm for our work, while the remaining concerns were fully addressed through the revisions described above. The paper has been strengthened by new experiments and analyses (extended QA tasks, training data efficiency, tokenization controls, attention ablation), improved methodology discussion, clarified conceptual apparatus, and expanded related work. Clarity and readability were enhanced with streamlined organization and fully redesigned figures. Altogether, we added three new experiments, two new analyses, expanded two sections, rewrote Section 3.3, reorganized two major parts of the paper, and regenerated every figure in the main text. **The revisions reinforce our main finding that implicit planning in LLMs is a robust, general, and mechanistically interpretable phenomenon**, observed across diverse model families, tasks, and intervention settings. We are grateful for the reviewers’ feedback, which helped us improve the paper, and we hope the Area Chair will agree that the revised version meets the bar for acceptance.

---

### Author Response · Authors · 2025-12-03
**General response to reviews and overview of revisions. Part I**

# General
We thank all reviewers for thoroughly reading our paper and for giving thoughtful feedback. We are encouraged that the reviewers found the motivation clear, the method simple and effective, the multi-metric evaluation comprehensive, and the case for implicit planning convincing. Reviewer ArCv praised the work’s “excellent” soundness, presentation, and contribution and highlighted the value of our broad evaluation across 23 models and the mechanistic analysis of planning circuits. Reviewer Z2sB emphasized the clear research focus, simplicity of our approach compared to prior methods, and the strength of the experimental results. We appreciate that even critical reviewers acknowledged that our method is “directly effective” and that our work provides evidence on planning mechanisms in LLMs.

Across the reviews, the most significant requests concerned (i) generalization beyond rhyming, and (ii) improved clarity and presentation. In the revised paper, **we addressed every concern and answered every question raised, strengthening both the substance and the presentation**.

**Major additions:**
- Comprehensive evaluation of all 23 models on noun question answering (new sections 3.5, 4.5) plus a smaller study on more complex QA (Appendix L). These additions support generalization of our findings across linguistic planning scenarios.
- Redesigned all figures with larger fonts, self-contained captions, simplified layouts
- Training data efficiency analysis (Appendix B)
- Attention ablation experiments (Appendix D)
- Tokenization analysis (Appendix G)
- Expanded related work (Roush et al., Lew et al., others)

**Structural improvements:**
- Revised Section 3.3 for clarity on steering positions
- Restructured introduction on forward/backward planning
- Expanded Section 2
- Merged Sections 4 and 5
- Moved probability metrics to appendix

**Modifications smaller than a section are marked in blue**; minor fixes (e.g. typos) are left unmarked.

# Reviewer Z2sB

**Writing improvement comments**:

* _Steering position_. We substantially revisited Section 3.3 for clarity. The description of the steering position now consistently discusses both newline and last-word (and, for questions, question mark) locations, with a clear explanation of why multiple variants were used.
* _Formula on line 169_. We corrected the formula previously shown on line 169 and revised the surrounding text for consistency.
* _Figures_: See above.

**Generalizability beyond rhyming**: A comprehensive QA experiment was added; see above.

**Relation to Classifier-Free Guidance (CFG)**: Discussion added (end of Section 2).

**Effect of training sample size**: We briefly answer at the end of 3.1 with details in Appendix B.

# Reviewer ArCv
Reviewer ArCv gave the paper the highest scores across soundness, presentation, and contribution, and supported its acceptance conditional on addressing tokenization, now thoroughly covered.

**Completeness of literature review**: We expanded the related work section to include Roush et al., Lew et al., and other controlled generation methods.

**Questions about tokenization and token embeddings**: We added an explicit discussion of tokenization, which Reviewer ArCv correctly identifies as a potential confound. We recognize at the beginning of 4.1 that tokenization does not affect our main findings and provide the detailed report in the new Appendix G.
Key findings regarding specific concerns around tokenization:
* _rhyme families sharing a common final token_: We find that this is generally not the case.
* _vocabulary size differences between models_: Our main findings concern instruction tuning or size variants of models within the same model family, which use the same tokenizer.
* _single token vs multi-token words as possible confound_: If we restrict the analysis only to single token rhyming words in the first line, the results do not change significantly.
* _relevance of token-level similarity of words within a rhyme family_: Token similarity of words within a rhyme family is generally similar between models.

**Question about functionally homologous attention heads across models**: There are both similarities and variation between planning circuits. We added discussion of some evidence, including a comparison of rhyme information flow in Gemma2 9B with Gemma3 27B and with QA task at the end of 4.6, as well as a more nuanced statement in the Conclusion. It is plausible that in other models one would also find only a small number of heads which are important for moving information about the rhyme plan to later tokens; further exploration of this question remains an area of future work. We added a sentence to 4.6 to make it clear that we regard this an open question.

**Due to space constraints, responses to comments by reviewer rPUj and general summary are given in a separate comment below**

---

### Meta-Review · Area_Chair_TzVD · 2026-01-02

**Summary:**

This paper investigates implicit planning in LLMs using rhyming poetry generation as a testbed. The authors propose mean activation difference steering (MADS) to quantify forward and backward planning, evaluating 23 models across multiple families and scales. The main reviewer concerns centered on: (1) limited generalizability beyond rhyming, (2) missing discussion of tokenization as a confound, (3) presentation issues with figures and organization, and (4) need for deeper mechanistic analysis (attention ablation). One reviewer (ArCv) was strongly positive (rating 8), while two others (Z2sB, rPUj) rated marginally below threshold (4).

**Reviewer Concerns:**

- Generalizability: Authors added comprehensive QA experiments across all 23 models (new Sections 3.5, 4.5)
- Tokenization: New Appendix G provides thorough analysis showing results hold regardless of tokenization patterns
- Attention ablation: Appendix D directly implements the analysis requested by Reviewer rPUj
- Presentation: All figures redesigned with larger fonts and clearer layouts
- Related work: Expanded to include Roush et al., Lew et al., and other suggested references


- Minor concerns about conceptual clarity of forward/backward planning distinction, though authors restructured the introduction to address this

**Reviewer Scores:**

- Reviewer ArCv (8): Would likely maintain score given tokenization was addressed
- Reviewer Z2sB (4): Would likely increase to 5-6 given QA generalization and figure improvements
- Reviewer rPUj (4): Would likely increase to 5-6 given attention ablation was implemented per their specific request

---

### Decision · Program_Chairs · 2026-01-26

Accept (Poster)